# Efficient High-Dimensional Data Representation Learning via Semi-Stochastic Block Coordinate Descent Methods

## Abstract

With the increase of data volume and data dimension, sparse representation learning attracts more and more attention. For high-dimensional data, randomized block coordinate descent methods perform well because they do not need to calculate the gradient along the whole dimension. Existing hard thresholding algorithms evaluate gradients followed by a hard thresholding operation to update the model parameter, which leads to slow convergence. To address this issue, we propose a novel hard thresholding algorithm, called Semi-stochastic Block Coordinate Descent Hard Thresholding Pursuit (SBCD-HTP). Moreover, we present its sparse and asynchronous parallel variants. We theoretically analyze the convergence properties of our algorithms, which show that they have a significantly lower hard thresholding complexity than existing algorithms. Our empirical evaluations on real-world datasets and face recognition tasks demonstrate the superior performance of our algorithms for sparsity-constrained optimization problems.

## 1 Introduction

In modern high-dimensional data analytics, where the variable dimension can be equal or even larger than the number of samples, sparse representation learning has become a mainstream method to explore the potential real model of the problem and provides statistically reliable results. It has been applied to many diverse domains such as high-dimensional statistics (Bühlmann & Van De Geer, 2011), signal processing (Lai et al., 2013), and computer vision (Wright et al., 2008). Many sparse learning algorithms such as $\ell_1$-norm convex relaxation methods (Negahban et al., 2009; Van de Geer et al., 2008) have been proposed in the past few decades. In order to get a smaller estimation error, $\ell_0$-norm constrained algorithms are more prominent than $\ell_1$-norm convex relaxation algorithms (Zhang et al., 2010). In this paper, we mainly focus on the following sparsity-constrained optimization problem:

$$\min_{w} \ F(w) := \frac{1}{n} \sum_{i=1}^{n} f_i(w), \quad \text{s.t.,} \ \|w\|_0 \le s, \tag{1}$$

where $F(w)$ is a finite-sum convex and smooth function, each function $f_i(w)$ is associated with the $i$-th sample, $\|w\|_0$ is the number of nonzero entries in variable $w$, and $s$ represents the sparsity level. The goal of sparse representation learning is to recover $w^*$ based on the given data. Such a formulation encapsulates plentiful important problems, including sparse graphical model learning (Zhou et al., 2019), sparse linear/logistic regression (Blumensath & Davies, 2009; Foucart, 2011; Pati et al., 1993; Bahmani et al., 2013), and low-rank regression (Rohde et al., 2011).

However, Problem (1) is in general NP hard, which is caused by the non-convexity of the sparsity constraint. It makes us to obtain an approximate solution to Problem (1). One of the most widely used methods for solving this problem is the hard thresholding based gradient descent method. In recent years, sparse representation learning is becoming more common for large-scale and high-dimensional data. However, deterministic gradient descent hard thresholding algorithms such as fast gradient descent hard thresholding (FG-HT) (Jain et al., 2014; Yuan et al., 2014) have to compute the gradients of all $n$ component functions, which leads to a huge computing overhead. To address this issue, many stochastic hard thresholding methods have been proposed. For instance, Nguyen et al. (2017) proposed a stochastic gradient descent hard thresholding (SG-HT) algorithm, whose

Table 1: Comparison of some properties of the hard thresholding algorithms for solving sparsity-constrained problems. $|\mathcal{B}|$ is the mini-batch size of block coordinate descent methods, and $k$ is the number of coordinate blocks. Since the sparsity level $s = \Omega(\kappa_{\widehat{s}} s^*)$ required in our algorithms (i.e., SBCD-HTP and its parallel variant, ASBCD-HTP) is much smaller than those of other algorithms (e.g., $\Omega(\kappa_{\widetilde{s}}^2 s^*)$), the condition number $\kappa_{\widehat{s}}$ in our algorithms is also smaller than $\kappa_{\widetilde{s}}$ in other algorithms, where $\widehat{s} = 2\Omega(\kappa_{\widehat{s}} s^*) + s^*$ and $\widetilde{s} = 2\Omega(\kappa_{\widetilde{s}}^2 s^*) + s^*$. $\widehat{\mathcal{I}} = \text{supp}(HT(\nabla F(w^*), 2s)) \cup \text{supp}(w^*)$ with $s = \Omega(\kappa_{\widetilde{s}}^2 s^*)$ is a support set.

| Methods | Required value of $s$ | Gradient Complexity | Hard Thresholding Complexity | Statistical Error |
|---|---|---|---|---|
| FG-HT | $\Omega(\kappa_{\widetilde{s}}^2 s^*)$ | $\mathcal{O}(n\kappa_{\widetilde{s}}\log(\frac{1}{\epsilon}))$ | $\mathcal{O}(\kappa_{\widetilde{s}}\log(\frac{1}{\epsilon}))$ | $\mathcal{O}(\sqrt{\Omega(\kappa_{\widetilde{s}}^2 s^*) + s^*}\|\nabla F(w^*)\|_\infty)$ |
| SG-HT | $\Omega(\kappa_{\widetilde{s}}^2 s^*)$ | $\mathcal{O}(\kappa_{\widetilde{s}}\log(\frac{1}{\epsilon}))$ | $\mathcal{O}(\kappa_{\widetilde{s}}\log(\frac{1}{\epsilon}))$ | $\mathcal{O}(\frac{1}{n}\sum_{i=1}^n \|\nabla f_i(w^*)\|_2)$ |
| SVRG-HT | $\Omega(\kappa_{\widetilde{s}}^2 s^*)$ | $\mathcal{O}((n+\kappa_{\widetilde{s}})\log(\frac{1}{\epsilon}))$ | $\mathcal{O}(\kappa_{\widetilde{s}}\log(\frac{1}{\epsilon}))$ | $\mathcal{O}(\sqrt{\widetilde{s}}\|\nabla F(w^*)\|_\infty + \|\nabla_{\widehat{\mathcal{I}}} F(x^*)\|)$ |
| ASBCDHT | $\Omega(\kappa_{\widetilde{s}}^2 s^*)$ | $\mathcal{O}((n+\frac{\kappa_{\widetilde{s}}|\mathcal{B}|}{k})\log(\frac{1}{\epsilon}))$ | $\mathcal{O}(\kappa_{\widetilde{s}}\log(\frac{1}{\epsilon}))$ | $\mathcal{O}(\sqrt{\widetilde{s}}\|\nabla F(w^*)\|_\infty)$ |
| SBCD-HTP | $\Omega(\kappa_{\widehat{s}} s^*)$ | $\mathcal{O}((n+\frac{\kappa_{\widehat{s}}|\mathcal{B}|}{k})\log(\frac{1}{\epsilon}))$ | $\mathcal{O}(\log(\frac{1}{\epsilon}))$ | $\mathcal{O}(\sqrt{\widehat{s}}\|\nabla F(w^*)\|_\infty)$ |
| ASBCD-HTP | $\Omega(\kappa_{\widehat{s}} s^*)$ | $\mathcal{O}((n+\frac{\kappa_{\widehat{s}}}{k})\log(\frac{1}{\epsilon}))$ | $\mathcal{O}(\log(\frac{1}{\epsilon}))$ | $\mathcal{O}(\sqrt{\widehat{s}}\|\nabla F(w^*)\|_\infty)$ |
| S$^2$BCD-HTP | $\Omega(\kappa_{\widehat{s}} s^*)$ | $\mathcal{O}((n+\frac{\kappa_{\widehat{s}}}{k})\log(\frac{1}{\epsilon}))$ | $\mathcal{O}(\log(\frac{1}{\epsilon}))$ | $\mathcal{O}(\sqrt{\widehat{s}}\|\nabla F(w^*)\|_\infty)$ |

gradient and hard thresholding complexities are both $\mathcal{O}(\kappa_{\widetilde{s}}\log(\frac{1}{\epsilon}))$. However, due to the stochastic sampling, SG-HT can only attain a sub-optimal estimation bound, as shown in Table 1, which is inferior to those of deterministic gradient methods such as FG-HT. Another limitation of SG-HT is that it requires that the restricted condition number $\kappa_{\widetilde{s}}$ should not be larger than $4/3$, which makes SG-HT hard for solving high-dimensional representation learning problems.

Recently, many stochastic variance reduced methods (e.g., SAG (Roux et al., 2012), SVRG (Johnson & Zhang, 2013)) and their accelerated variants such as (Defazio, 2016; Allen-Zhu, 2018) have been proposed to accelerate stochastic gradient methods for convex optimization. All these methods enjoy low per-iteration complexities comparable with stochastic gradient descent (SGD), and they also can attain improved convergence rates. By incorporating the variance reduction technique into sparsity-constrained optimization domain, Li et al. (2016b) proposed a stochastic variance reduced gradient hard thresholding (SVRG-HT) algorithm, which can converge more quickly and obtain a smaller estimation error than SG-HT. Moreover, SVRG-HT allows an arbitrarily large condition number in its theoretical analysis similar to FG-HT (Yuan et al., 2014). The gradient oracle and hard thresholding complexities of SVRG-HT are $\mathcal{O}((n+\kappa_{\widetilde{s}})\log(\frac{1}{\epsilon}))$ and $\mathcal{O}(\kappa_{\widetilde{s}}\log(\frac{1}{\epsilon}))$, respectively. Nevertheless, it can not leverage the coordinate block to accelerate convergence. Chen & Gu (2016) proposed an accelerated stochastic block coordinate gradient descent hard thresholding (ASBCDHT) algorithm. The gradient oracle complexity of ASBCDHT is $\mathcal{O}((n+\frac{\kappa_{\widetilde{s}}|\mathcal{B}|}{k})\log(\frac{1}{\epsilon}))$, which is superior to SVRG-HT. Moreover, ASBCDHT also has the hard thresholding complexity, $\mathcal{O}(\kappa_{\widetilde{s}}\log(\frac{1}{\epsilon}))$. However, the hard thresholding complexity of ASBCDHT still scales linearly with $\kappa_{\widetilde{s}}$, which is usually expensive for real-world sparse learning problems. For large-scale and high-dimensional data, Li et al. (2016a) also proposed the asynchronous parallel variant of SVRG-HT (called ASVRG-HT) by utilizing multicore architectures. Although it makes each processor to evaluate a stochastic gradient update on a global parameter stored in a shared memory in an asynchronous and lock-free mode, ASVRG-HT attains the similar gradient and hard thresholding complexities as its general version (i.e., SVRG-HT). Therefore, all the algorithms mentioned above have a linearly $\kappa_{\widetilde{s}}$-dependent hard thresholding complexity. This motivates us to address the following key issue:

**Can we design such algorithms that have a $\kappa_{\widetilde{s}}$-independent hard thresholding complexity and a lower gradient oracle complexity?**

To answer the above problem, we propose an efficient Semi-stochastic Block Coordinate Descent Hard Thresholding Pursuit (SBCD-HTP) algorithm and its sparse Asynchronous variant (ASBCD-HTP). The oracle complexities and statistical estimation error of the proposed algorithms and other hard thresholding methods are summarized in Table 1. We highlight several theoretical advantages of the proposed algorithms over the state-of-the-art methods as follows:

• SBCD-HTP and ASBCD-HTP substantially improve the restricted condition on the sparsity level $s$, i.e., $s = \Omega(\kappa_{\widehat{s}} s^*)$, while most related algorithms such as SVRG-HT require $s = \Omega(\kappa_{\widetilde{s}}^2 s^*)$.

• The statistical estimation error of both our algorithms is better than those of FG-HT, SVRG-HT and ASBCDHT, as our algorithms have a smaller sparsity level $s = \Omega(\kappa_{\widehat{s}} s^*)$ and thus have a smaller cardinality, i.e., $\widehat{s} = 2s + s^*$. This makes the restricted condition number $\kappa_{\widehat{s}}$ of SBCD-HTP smaller than other algorithms. Moreover, the statistical error of our algorithms is also smaller than that of SG-HT (i.e., $\frac{1}{n}\sum_{i=1}^{n}\|\nabla f_i(w^*)\|$) due to the large magnitude of $\|\nabla f_i(w^*)\|$ (Zhou et al., 2018b).

• Both SBCD-HTP and ASBCD-HTP enjoy a $\kappa_{\widehat{s}}$-independent hard thresholding complexity, which is significantly better than the state-of-the-art hard thresholding algorithms. That is, the hard thresholding complexity of our algorithms is $\kappa_{\widetilde{s}}$ times lower than that of FG-HT, SG-HT, SVRG-HT and ASBCDHT with $\widetilde{s} = 2\Omega(\kappa_{\widetilde{s}}^2 s^*) + s^*$. Since both our algorithms have a significantly lower hard thresholding complexity, they are more suitable for handling large-scale sparse representation learning problems, especially for high-dimensional data.

• For sparsity-constrained problems, the gradient oracle complexities of both SBCD-HTP and ASBCD-HTP (or S²BCD-HTP) are much lower than those of the state-of-the-art hard thresholding algorithms, and that of S²BCD-HTP is similar to ASBCD-HTP under conditions on the delay. Actually, the gradient oracle complexity of S²BCD-HTP is better than SBCD-HTP when we deal with sparse data sets, since S²BCD-HTP only evaluates the common nonzero point of the random sample and the sampled block. Furthermore, their gradient oracle complexities are significantly lower than those of ASBCDHT and SVRG-HT, and much lower than that of FG-HT, since they require a smaller sparsity level $s = \Omega(\kappa_{\widehat{s}} s^*)$ and also have a smaller value of $\kappa_{\widehat{s}}$ with $\widehat{s} = 2s + s^*$ compared with the restricted condition number $\kappa_{\widetilde{s}}$ used in other hard thresholding algorithms (Zhou et al., 2018b).

## 2 RELATED WORK

In this section, we briefly discuss the relevant research to our work, which beyonds the sparsity-constrained optimization domain. Traditional gradient descent is computational expensive at each iteration, and stochastic gradient descent has a low per-iteration complexity while obtaining a large variance in estimating. Thus, many stochastic variance reduced algorithms (Defazio et al., 2014; Roux et al., 2012; Shalev-Shwartz & Zhang, 2013; Shang et al., 2018; Johnson & Zhang, 2013) and their variants (Schmidt et al., 2017; Konečný et al., 2015; Xiao & Zhang, 2014; Hannah et al., 2018) have been proposed. The stochastic variance reduced methods are very promising for many machine learning problems including sparse learning problems.

In contrast to gradient descent methods, coordinate descent algorithms have received increasing attention due to their successful applications in high dimensional problems (Breheny & Huang, 2011; Friedman et al., 2007). Among them, randomized block coordinate descent (RBCD) updates a block of coordinate with respect to the entire training instances. The per-iteration cost is significantly lower than gradient descent, while it still has a relationship with $n$ component functions. Some stochastic block coordinate descent (SBCD) algorithms such as (Dang & Lan, 2015; Xu & Yin, 2015; Konečný et al., 2017) were proposed. Such algorithms compute the stochastic gradient with respect to a random sample restricted to one randomized coordinate block. Therefore, these algorithms randomly sample a block of features and data instances at each iteration. However, they can only achieve a sublinear convergence rate (Zhang & Gu, 2016). Recently, some mini-batch randomized block coordinate descent algorithms such as (Zhao et al., 2014; Wang & Banerjee, 2014) were proposed to accelerate the convergence of stochastic block coordinate gradient descent. Our work studies over the above research by considering a sparsity-constrained optimization problem. The proposed algorithms enjoy a lower computational complexity in both gradient evaluation and hard thresholding computation while obtaining a linear convergence performance. Moreover, its sparse and asynchronous variants are also proposed to deal with sparse high-dimensional data.

## 3 PRELIMINARIES

Throughout this paper, we use $w^*$ to denote the optimal solution of Problem (1), and its optimal sparsity level is $s^*$ with $\|w^*\|_0 \le s^*$. $\|w\|$ is the Euclidean norm for a vector $w \in \mathbb{R}^d$ and $\|w\|_\infty$ is the largest absolute entry in $w$. The hard thresholding operation $HT(w, s)$ preserves the $s$ largest entries of $w$ in magnitude for vector $w$ and the rest entries are set to zero. We use supp$(w)$ to denote the support set of $w$, i.e., the indices of its nonzero entries. Given an index set $\mathcal{I}$, we define $\mathcal{I}^C$ as the complement set of $\mathcal{I}$, and $v_{\mathcal{I}} \in \mathbb{R}^d$, where $[v_{\mathcal{I}}]_j = v_j$ if $j \in \mathcal{I}$ and $[v_{\mathcal{I}}]_j = 0$ if $j \notin \mathcal{I}$. Given an integer

---

**Algorithm 1** Semi-stochastic Block Coordinate Descent Hard Thresholding Pursuit (SBCD-HTP)

---

**Input:** Number of outer-loops $R$, number of inner-loops $m$, step size $\eta$, sparsity level $s$.
**Initialize:** $\widetilde{w}^0$.
1: **for** $r = 0, 1, \ldots, R - 1$ **do**
2:     $w^0 = \widetilde{w} = \widetilde{w}^r, \;\; \nabla F(\widetilde{w}) = \frac{1}{n} \sum_{i=1}^{n} \nabla f_i(\widetilde{w}), \;\; \widetilde{\mathcal{G}} = \text{supp}(\widetilde{w});$
3:     **for** $t = 0, 1, \ldots, m - 1$ **do**
4:         Randomly sample a mini-batch $\mathcal{B}$ from $[n]$ uniformly;
5:         Randomly sample a block $j_t$ from $[k]$ uniformly, and $\mathcal{S} = \widetilde{\mathcal{G}} \cup \mathcal{G}_{j_t}$;
6:         $\nabla_S g(w^t) = \frac{1}{|\mathcal{B}|} \sum_{i_t \in \mathcal{B}} \nabla_S f_{i_t}(w^t) - \nabla_S f_{i_t}(\widetilde{w}) + \nabla_S F(\widetilde{w});$
7:         $w^{t+1} = w^t - \eta \nabla_S g(w^t);$
8:     **end for**
9:     $\widetilde{w}^{r+1} = HT(w^m, s);$
10: **end for**
**Output:** $\widetilde{w}^R$.

---

$n \geq 1$, we define $[n] = \{1, \cdots, n\}$. For a set $\mathcal{B}$, we denote its cardinality by $|\mathcal{B}|$. Moreover, we use the common notations of $\Omega(\cdot)$ and $\mathcal{O}(\cdot)$ to characterize the asymptotics of two real sequences.

Throughout the analysis, we make two important assumptions on the objective function, which are commonly used in the analysis of hard thresholding algorithms (Li et al., 2016b; Shen & Li, 2017; Chen & Gu, 2016; Gao & Huang, 2018). In high-dimensional sparse learning, the per-iteration hard thresholding operation can be time-consuming or even more expensive than gradient computation. Thus, we also take the complexity of hard thresholding into our consideration.

**Assumption 1 (Restricted Strong Convexity)** *A differentiable function $F(\cdot)$ is restricted $\rho_{\widehat{s}}^-$-strongly convex at sparsity level $\widehat{s}$, if there exists a uniform constant $\rho_{\widehat{s}}^- > 0$ such that for any $w, w' \in \mathbb{R}^d$ with $\|w - w'\|_0 \leq \widehat{s}$, we have*

$$F(w) - F(w') - \langle \nabla F(w'), \, w - w' \rangle \geq \frac{\rho_{\widehat{s}}^-}{2} \|w - w'\|^2. \tag{2}$$

**Assumption 2 (Restricted Strong Smoothness)** *For any $i \in [n]$, the differentiable function $f_i(\cdot)$ is restricted $\rho_{\widehat{s}}^+$-strongly smooth at sparsity level $\widehat{s}$, if there exists a uniform constant $\rho_{\widehat{s}}^+ > 0$ such that for any $w, w' \in \mathbb{R}^d$ with $\|w - w'\|_0 \leq \widehat{s}$, we have*

$$f_i(w) - f_i(w') - \langle \nabla f_i(w'), \, w - w' \rangle \leq \frac{\rho_{\widehat{s}}^+}{2} \|w - w'\|^2. \tag{3}$$

Moreover, the restricted condition number is defined as: $\kappa_{\widehat{s}} = \rho_{\widehat{s}}^+ / \rho_{\widehat{s}}^-$. Since the complexity of one hard thresholding operation is linear with $d$, which is similar to a computational cost of a stochastic gradient, we define the hard thresholding complexity as a metric for hard thresholding algorithms.

**Definition 1 (Hard Thresholding Complexity)** *In a hard thresholding operation, a vector $w$ is fed into $HT(\cdot, s)$ and then the output $HT(w, s)$ is obtained.*

Both gradient oracle and hard thresholding complexities can more comprehensively reflect the overall computational cost of first-order hard thresholding algorithms. Therefore, their per-iteration cost is dominated by both the gradient evaluation and hard thresholding operations.

We also define the optimization error and statistical error to clearly understand our theoretical results.

**Definition 2 (Optimization Error and Statistical Error)** *The optimization error is the difference which will decrease with the increasing of iterations. A statistical error is the (unknown) difference between the retained value and the true value.*

This means that the optimization error will decrease to zero when the number of iteration is quite large. However, a statistical error will still work during the optimized process. Therefore, the larger the statistical error, the more difference to the true value.

## 4 SEMI-STOCHASTIC BLOCK COORDINATE DESCENT HARD THRESHOLDING PURSUIT

In this section, we propose a novel Semi-stochastic Block Coordinate Descent Hard Thresholding Pursuit (SBCD-HTP) algorithm, and also theoretically analyze its convergence properties.

### 4.1 OUR SBCD-HTP ALGORITHM

The proposed SBCD-HTP algorithm is summarized in Algorithm 1. In each outer-loop, we select a snapshot point $\widetilde{w}$ and compute its full gradient $\nabla F(\widetilde{w})$. In each inner-loop, we uniformly and randomly select a mini-batch samples $\mathcal{B}$ and a block of coordinates $\mathcal{G}_{j_t}$, where $j_t \in \{1, 2, \ldots, k\}$, $\{\mathcal{G}_1 \ldots \mathcal{G}_k\}$ is a partition of all the $d$ coordinates and is divided uniformly at random. We usually set $k = 10$ in our experiments. Note that, different from ASBCDHT, in line 5 we set $\mathcal{S}$ to be a union set of the support set of snapshot point $\widetilde{w}$ and $\mathcal{G}_{j_t}$. In this way, the coordinate of the support set $\mathcal{S}$ which includes the possibly optimal coordinates can be updated, and thus this can reduce the error caused by randomness. Moreover, SBCD-HTP enjoys the dual advantages of randomized block coordinate descent and semi-stochastic gradient descent to optimize the objective as shown in line 6. Note that we perform the hard thresholding operation in only the outer-loop, which is different from all existing hard thresholding algorithms. Performing hard thresholding operations with high frequency or prematurely in inner-loop will make the gradient information lost, which is the main reason for slow convergence and low accuracy of existing hard thresholding sparsity-constrained algorithms. Our hard thresholding algorithm not only guarantees the sparsity of model parameters, but also uses the full dimensional information without the interference of hard thresholding operations to update model parameter $w^{t+1}$. Our theoretical analysis and experimental results show that SBCD-HTP has a faster convergence rate and more accurate results.

We have to clarify the difference between the proposed SBCD-HTP and ASBCDHT in Chen & Gu (2016) for clear improvements. There are mainly two changes: a) we design a bigger support set with the union of the sampled block $\mathcal{G}_{j_t}$ with the one of the current support of the snapshot point $\widetilde{w}$, since the current support of snapshot point contain the most related optimized information, while that of Chen & Gu (2016) only use the sampled block. b) we move the hard thresholding operation outside of the inner-loop, since the truncated function (hard thresholding) may loss many information and bringing higher complexity. Moreover, the asynchronous variant of SBCD-HTP is also proposed in this paper to optimize the sparse problem 1 in the sparse data. This is the first to design this type of algorithm in sparse learning field, and the theoretical proof is also provided.

### 4.2 CONVERGENCE ANALYSIS

We first analyze the convergence behavior of SBCD-HTP. The main result is summarized in the following theorem, whose proof is provided in Section C.

**Theorem 1** *Suppose $F(w)$ is $\rho_{\widehat{s}}^-$-strongly convex and each function $f_i(w)$ is $\rho_{\widehat{s}}^+$-strongly smooth with parameter $\widehat{s} = 2s + s^*$. Let $\kappa_{\widehat{s}} = \frac{\rho_{\widehat{s}}^+}{\rho_{\widehat{s}}^-}$ and the sparsity level $s \geq \Omega(\kappa_{\widehat{s}} s^*)$. In addition, assume that the learning rate $\eta \leq \frac{1}{48\rho_{\widehat{s}}^+}$, the number of inner-loops $m \geq 1800\kappa_{\widehat{s}}$, and the number of blocks $k = 10$, then the convergence rate $\gamma = \frac{k}{2\eta m \rho_{\widehat{s}}^-(1-\omega)} + \frac{8\rho_{\widehat{s}}^+ \eta(n-|\mathcal{B}|)}{|\mathcal{B}|(1-\omega)(n-1)} \leq \frac{1}{2}$. For the sparsity-constrained problem (1), the output of Algorithm 1 satisfies*

$$\mathbb{E}[F(\widetilde{w}^r) - F(w^*)] \leq (\frac{1}{2})^r \mathbb{E}[F(\widetilde{w}^0) - F(w^*)] + \frac{\eta}{1-\omega}\|\nabla_{\tilde{\mathcal{I}}} F(w^*)\|^2, \tag{4}$$

*where $\tilde{\mathcal{I}} = supp(HT(\nabla F(w^*), 2s)) \cup supp(w^*)$ and $\omega = 8\rho_{\widehat{s}}^+ \eta(1 + \frac{n-|\mathcal{B}|}{|\mathcal{B}|(n-1)})$.*

Theorem 1 shows that SBCD-HTP enjoys a linear convergence rate for solving sparsity-constrained problem (1). In order to ensure that the linear converging term in Eq. (4) satisfies $(\frac{1}{2})^r \mathbb{E}[F(\widetilde{w}^0) - F(w^*)] \leq \epsilon$, we can obtain the following corollary.

**Corollary 1** *Suppose the conditions in Theorem 1 hold. To achieve $(\frac{1}{2})^r \mathbb{E}[F(\widetilde{w}^0) - F(w^*)] \leq \epsilon$, the gradient oracle complexity of SBCD-HTP is $\mathcal{O}((n + \frac{\kappa_{\widehat{s}}|\mathcal{B}|}{k})\log(\frac{1}{\epsilon}))$, and its hard thresholding complexity is $\mathcal{O}(\log(\frac{1}{\epsilon}))$.*

**Remark 1** *This corollary is inferred under the high-dimensional condition. Actually, the worst condition of the gradient oracle complexity of SBCD-HTP is $\mathcal{O}((n + \kappa_{\widehat{s}}|\mathcal{B}|^{\frac{d+s}{d}})\log(\frac{1}{\epsilon}))$. Due to the small sparsity level $s$ required in our Theorem 1 and the overlaps between $\mathcal{G}_{j_t}$ and $\widetilde{\mathcal{G}}$, we can obtain the gradient oracle complexity $\mathcal{O}((n + \frac{\kappa_{\widehat{s}}|\mathcal{B}|}{k})\log(\frac{1}{\epsilon}))$ for SBCD-HTP.*

Table 1 summarizes the properties of some hard thresholding methods. Compared with the gradient oracle complexities of FG-HT (Yuan et al., 2014) and SVRG-HT (Li et al., 2016b) (i.e., $\mathcal{O}(n\kappa_{\widetilde{s}}\log(\frac{1}{\epsilon}))$ and $\mathcal{O}((n + \kappa_{\widetilde{s}})\log(\frac{1}{\epsilon}))$, respectively), SBCD-HTP has a much lower oracle complexity. For ASBCDHT (Chen & Gu, 2016), whose gradient oracle complexity is $\mathcal{O}((n + \frac{\kappa_{\widetilde{s}}|\mathcal{B}|}{k})\log(\frac{1}{\epsilon}))$, the gradient oracle complexity of SBCD-HTP is $\mathcal{O}((n + \frac{\kappa_{\widehat{s}}|\mathcal{B}|}{k})\log(\frac{1}{\epsilon}))$ and is much lower than AS-BCDHT, as $\kappa_{\widehat{s}}$ of SBCD-HTP is usually comparable to or even smaller than those of others (note that $s$ in $\widehat{s} = 2s + s^*$ is required to be smaller than others as shown below). Moreover, SBCD-HTP allows $s = \Omega(\kappa_{\widehat{s}}s^*)$, which is considerably superior to the condition of $s = \Omega(\kappa_{\widetilde{s}}^2 s^*)$ required in other hard thresholding algorithms such as (Zhou et al., 2018b). In particular, with the same parameter settings of $|\mathcal{B}|$ and $k$, the gradient oracle complexity of our algorithm outperforms ASBCDHT. SBCD-HTP also attains a lower hard thresholding complexity than the state-of-the-art hard thresholding methods. That is, the hard thresholding complexity of SBCD-HTP is $\mathcal{O}(\log(\frac{1}{\epsilon}))$, which is $\kappa_{\widetilde{s}}$-independent and is significantly lower than those of other methods. This is a significant improvement on hard thresholding complexity. Theorem 1 immediately implies the following results.

**Corollary 2** *Suppose the conditions in Theorem 1 hold. The output of Algorithm 1 satisfies*

$$\mathbb{E}\|\widetilde{w}^r - w^*\| \leq \sqrt{\frac{2(\frac{1}{2})^r[F(\widetilde{w}^0) - F(w^*)]}{\rho_{\widehat{s}}^-}} + \left(\frac{2}{\rho_{\widehat{s}}^-} + \frac{2\eta}{1-\omega}\right)\sqrt{\widehat{s}}\|\nabla\mathcal{F}(w^*)\|_\infty. \tag{5}$$

The right-hand side of Eq. (5) consists of two terms. The first term is the optimization error, which approaches zero with the increase of $r$. The second term corresponds to the statistical error, which is proportional to $\sqrt{\widehat{s}}\|\nabla F(w^*)\|_\infty$. One can also observe that the statistical error bound is better than those of FG-HT and ASBCDHT, since $\widehat{s}$ in the statistical error $\mathcal{O}(\sqrt{\widehat{s}}\|\nabla F(w^*)\|_\infty)$ has a smaller cardinality (i.e., $2\Omega(\kappa_{\widehat{s}}s^*) + s^*$), while existing algorithms require a larger sparsity level $s' = \Omega(\kappa_{\widetilde{s}}^2 s^*)$ and have a larger cardinality. Therefore, compared with other existing algorithms, SBCD-HTP enjoys a smaller statistical error, especially for the problems with a large restricted condition number. It is usually better than the error bound $\mathcal{O}(\sqrt{\widetilde{s}}\|\nabla F(w^*)\|_\infty + \|\nabla_{\widetilde{\mathcal{I}}}F(w^*)\|)$ with $\widetilde{s} = 2\Omega(\kappa_{\widetilde{s}}^2 s^*) + s^*$ in SVRG-HT. Moreover, the error bound of SBCD-HTP is much smaller than that of SG-HT (Nguyen et al., 2017) (i.e., $\mathcal{O}(\frac{1}{n}\sum_{i=1}^n\|\nabla f_i(w^*)\|)$), since the magnitude of the individual gradient norm $\|\nabla f_i(w^*)\|$ can still be relatively large (Zhou et al., 2018b).

## 5 ASYNCHRONOUS VARIANT OF SBCD-HTP

In this section, we propose the serial sparse and asynchronous parallel variants of SBCD-HTP for high-dimensional sparse data sets. Our Asynchronous Sparse Semi-stochastic Block Coordinate Descent Hard Thresholding Pursuit (ASBCD-HTP) algorithm is summarized in Algorithm 2. The main difference between ASBCD-HTP and SBCD-HTP is sparse approximate gradients as in (Mania et al., 2017; Zhou et al., 2018a). In order to perform fully sparse updates, we use a diagonal matrix $D$ to re-weigh the dense fully gradient $\nabla F(\widetilde{w})$. The entries of $D$ are the inverse probabilities of the corresponding coordinates belonging to a uniformly sampled support $T_{i_t}$ of sample $i_t$. Let $P_{i_t}$ be the projection matrix for the support $T_{i_t}$, we can get $D_{i_t}$ since $D_{i_t} = P_{i_t}D$. Then we can find that $\mathbb{E}_{i_t}[D_{i_t}\nabla F(\widetilde{w})] = \nabla F(\widetilde{w})$. In this setting, we can compute the full gradient $\nabla F(\widetilde{w})$ also in a parallel way. In order to reduce thread interference and fully utilize the sparsity of datasets, we restrict the sparse gradient $\nabla g([\widehat{w}]_{T_{i_t}})$ to the intersection set of $\mathcal{S}$ and $T_{i_t}$ and obtain $\nabla_S g([\widehat{w}]_{T_{i_t}})$. Specifically, each thread of ASBCD-HTP computes sparse approximate gradient independently and then atomically write this gradient to the shared variable $\widetilde{w}^r$. Note that Sparse variant of SBCD-HTP ($S^2$BCD-HTP) can be viewed as the single-thread version of ASBCD-HTP, which has linear convergence and the same oracle complexity as ASBCD-HTP. The detailed description and analysis of $S^2$BCD-HTP is listed in Section D. Next we give the main theoretical result of ASBCD-HTP.

---

**Algorithm 2** Asynchronous Sparse SBCD-HTP (ASBCD-HTP)

---

**Input:** The number of outer-loops $R$, number of inner-loops $m$, step size $\eta$, sparsity level $s$.
**Initialize:** $\widetilde{w}^0$.
1: **for** $r = 0, 1, \ldots, R - 1$ **do**
2:    $w^0 = \widetilde{w} = \widetilde{w}^r$, $\widetilde{\mathcal{G}} = \text{supp}(\widetilde{w})$, and $\nabla F(\widetilde{w}) = \frac{1}{n}\sum_{i=1}^{n}\nabla f_i(\widetilde{w})$;   //computed in parallel
3:    $t = 0$;   //inner loop counter
4:    **compute the while loop in parallel**
5:    **while** $t < m$ **do**
6:       Randomly sample $i_t$ from $[n]$ uniformly, and randomly sample $j_t$ from $[k]$ uniformly;
7:       $T_{i_t}$ :=support of sample $i_t$, and $t = t + 1$;   //atomic increase counter $t$
8:       $\mathcal{S} = \widetilde{\mathcal{G}} \cup \mathcal{G}_{j_t}$, $[\widehat{w}]_{T_{i_t}} :=$ inconsistent read of shared variable $[\widetilde{w}^r]_{T_{i_t}}$;
9:       $\nabla_S g([\widehat{w}]_{T_{i_t}}) = \nabla_S f_{i_t}([\widehat{w}]_{T_{i_t}}) - \nabla_S f_{i_t}([\widetilde{w}]_{T_{i_t}}) + D_{i_t}\nabla_S F(\widetilde{w})$;
10:      $[\widetilde{w}^r]_{\mathcal{S}\bigcap T_{i_t}} = [\widetilde{w}^r]_{\mathcal{S}\bigcap T_{i_t}} - \eta\nabla_S g([\widehat{w}]_{T_{i_t}})$;   //atomic write
11:    **end while**
12:    $\widetilde{w}^{r+1} = HT(\widetilde{w}^r, s)$;
13: **end for**
**Output:** $\widetilde{w}^R$.

---

**Theorem 2** *Suppose $F(w)$ is $\rho_{\widehat{s}}^-$-strongly convex and each function $f_i(w)$ is $\rho_{\widehat{s}}^+$-strongly smooth with parameter $\widehat{s} = 2s + s^*$. Let $\kappa_{\widehat{s}} = \frac{\rho_{\widehat{s}}^+}{\rho_{\widehat{s}}^-}$ and the sparsity level $s \geq \Omega(\kappa_{\widehat{s}}s^*)$. We assume that the step size $\eta = \frac{1}{60\rho_{\widehat{s}}^+}$, the number of inner-loops $m \geq 120\kappa_{\widehat{s}}$, the number of blocks $k = 10$ and $\tau \leq \min\{\frac{3}{5\sqrt{\Delta}}, 2\kappa_{\widehat{s}}, \sqrt{\frac{2\kappa_{\widehat{s}}}{\sqrt{\Delta}}}\}$ (i.e., the linear speedup condition). Then for the sparsity-constrained problem (1), the gradient oracle and hard thresholding complexities of Algorithm 2 are*

$$\mathcal{O}\left((n + \kappa_{\widehat{s}}/k)\log(1/\epsilon)\right) \quad \text{and} \quad \mathcal{O}\left(\log(1/\epsilon)\right), \tag{6}$$

*where $\tau$ denotes the maximum number of concurrent threads (Mania et al., 2017) and $\Delta = \max_{j=1\cdots d} p_j$, which is a indicator to measure the sparsity of datasets (Leblond et al., 2017).*

Theorem 2 implies that ASBCD-HTP (or S$^2$BCD-HTP) can still obtain similar gradient oracle complexity and hard thresholding complexity as SBCD-HTP while obtaining a linear speedup ratio. Compared with ASVRG-HT (Li et al., 2016a), ASBCD-HTP has a much lower gradient complexity, i.e., $\mathcal{O}((n + \frac{\kappa_{\widehat{s}}}{k})\log\frac{1}{\epsilon})$ for ASBCD-HTP *vs.* $\mathcal{O}((n + \kappa_{\widetilde{s}})\log\frac{1}{\epsilon})$ for ASVRG-HT. For hard thresholding complexity, ASBCD-HTP is $\kappa_{\widetilde{s}}$ times lower than ASVRG-HT, which is a significant improvement in the asynchronous setting. Since ASVRG-HT has to compute hard thresholding operations at each inner-loop, the practical performance may be worse, especially for high-dimensional data. We also show this improvement by our empirical evaluations.

## 6 EXPERIMENTS

In this section, we evaluate the performance of SBCD-HTP, S$^2$BCD-HTP and ASBCD-HTP for solving sparse linear regression and sparse logistic regression on real-world datasets. All the dense algorithms were implemented in MATLAB, while the sparse and asynchronous algorithms were implemented in C++ and executed through MATLAB interface for a fair comparison. We also test the performance for face recognition tasks. The detailed information is described in Section F. Since there is no ground truth on real-world datasets, we run all these baselines sufficiently long until $\|w^r - w^{r+1}\|/\|w^r\| \leq 10^{-6}$, and then we use the minimum $F(w^r)$ as the approximate optimal value $F^*$ for sub-optimality estimation in our experimental results Zhou et al. (2018b).

### 6.1 DENSE ALGORITHMS

We ran the experiments of all the dense algorithms on a PC with an Intel i7-7700 CPU and 32GB RAM. We compare SBCD-HTP with several state-of-the-art sparsity-constrained algorithms, including FG-HT (Yuan et al., 2014), SG-HT (Nguyen et al., 2017), SVRG-HT (Li et al., 2016b), ASBCDHT (Chen & Gu, 2016) and FNHTP (Chen & Gu, 2017). The detailed descriptions of the

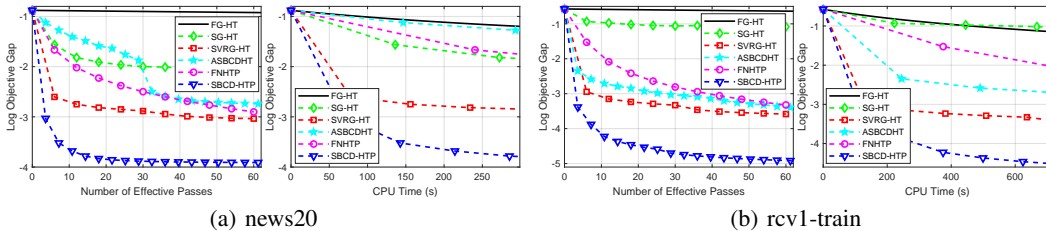

Figure 1: Comparison of all the hard thresholding algorithms for solving sparse logistic regression problems. In each plot, the vertical axis shows the objective value minus the minimum, and the horizontal axis is the number of effective passes over data (left) or running time (seconds, right).

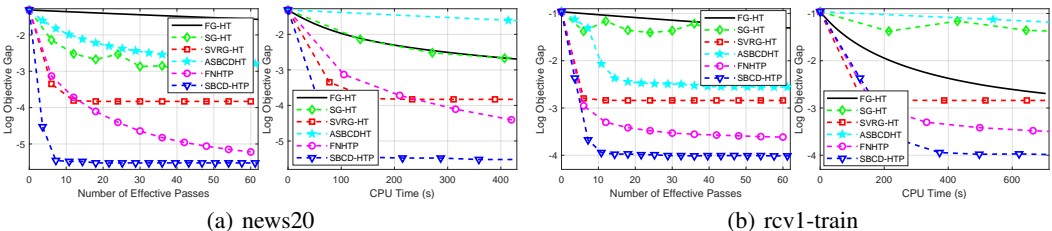

Figure 2: Comparison of all the hard thresholding algorithms for solving sparse linear regression problems.

algorithms and datasets are provided in Section F. We set the sparsity level $s = 200$ for real-word datasets. For the number of inner-loops, we set $m = 2n$ for SBCD-HTP and ASBCDHT, $m = n$ for SVRG-HT as suggested in (Li et al., 2016b; Chen & Gu, 2016) and 1000 (sparse linear regression)/3000 (sparse logistic regression) for FNHTP. We also set the batch-size of FNHTP to $s$, and that of ASBCDHT and SBCD-HTP to 5. For the block coordinate descent methods, we set the number of coordinate blocks $k$ to 10. All the algorithms are tuned to their best performance.

We report the performance of all the algorithms in terms of both effective passes and CPU time. More experimental results are presented in Section F. Figures 1 and 2 show that for both sparse linear regression and sparse logistic regression problems, our algorithm converges significantly faster than the baseline algorithms in terms of both effective passes and CPU time. The main reason is that we use all the information of gradients, while the related algorithms use less information caused by too frequent and premature hard thresholding operation in each inner-loop.

## 6.2 ASYNCHRONOUS ALGORITHMS

We ran the experiments of all the asynchronous algorithms on a PC with an Intel Xeon(R) Gold 5120 CPU and 64GB RAM. We compare our sparse and asynchronous algorithms with several state-of-the-art algorithms, including ASG-HT, ASVRG-HT and $A^2$SBCD-HT (asynchronous variant of ABSCDHT). For the algorithms, we set the number of threads to 20.

We only report the performance of all the algorithms in terms of CPU time since the performance in terms of effective passes is similar to the dense case. More experimental results are reported in Section F. Figure 3 shows that ASBCD-HTP and $S^2$BCD-HTP significantly outperform than other algorithms. Moreover, ASBCD-HTP has a faster convergence performance than $S^2$BCD-HTP because of asynchronous parallel acceleration. There are three main reasons for the advantages of our ASBCD-HTP: a) We get the set $\mathcal{S}$ from the support set of snapshot point $\widetilde{w}$ and $\mathcal{G}_{j_t}$. In this way, the coordinate of the support set which appears as the possible optimal solution can be updated, and thus reducing the error caused by randomness. b) Because we put the hard thresholding operation in each outer-loop, the computation cost per thread of our algorithm at each inner-loop is less, while each thread of the baseline algorithms has a hard thresholding operation, which is expensive especially for high-dimensional datasets. c) Our algorithm updates $\widetilde{w}^r$ with all information of gradients, while the baseline algorithms update $\widetilde{w}^r$ with less information caused by too frequent and premature hard thresholding operation in each inner-loop. We also evaluate the improvement of asynchronous parallel by running the same passes with different numbers of threads. We calculate the speed-up

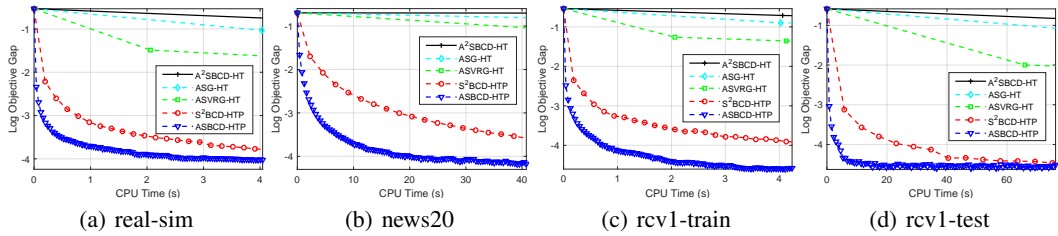

Figure 3: Comparison of ASG-HT, ASVRG-HT, A$^2$SBCD-HT, S$^2$BCD-HTP and ASBCD-HTP for solving sparse logistic regression problems. The four asynchronous parallel algorithms (i.e., ASG-HT, ASVRG-HT, A$^2$SBCD-HT and ASBCD-HTP) run on 20 threads.

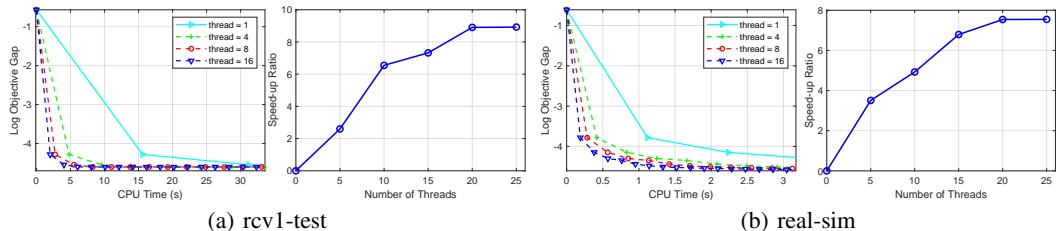

Figure 4: Speedup evaluation on rcv1-test and real-sim. Left: Evaluation of objective function gap in terms of CPU time for ASBCD-HTP with different threads. Right: Speedup ratio with respect to the number of threads.

ratio based on the running time of a single thread. From Figure 4, we can see that our ASBCD-HTP is accelerated by a nearly linear ratio.

## 7 CONCLUSION

In this paper, we proposed an efficient semi-stochastic block coordinate descent hard thresholding pursuit (SBCD-HTP) method for sparse representation learning. We proved that SBCD-HTP attains the gradient oracle complexity of $\mathcal{O}((n + \frac{\kappa_{\hat{s}}|\mathcal{B}|}{k}) \log(\frac{1}{\epsilon}))$, and the hard thresholding complexity of $\mathcal{O}(\log(\frac{1}{\epsilon}))$, respectively, which is an improvement over the existing algorithms. Moreover, we also presented the sparse and asynchronous parallel variants of SBCD-HTP. As far as we know, our algorithms are the first to use the full information of gradient to update. This is an improvement on sparse representation learning because it can be easily extended to other sparse learning algorithms.

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

APPENDIX

## A    KEY LEMMAS

**Lemma 1 ((Jain et al., 2014))** *For any index set $I$, any $w \in \mathbb{R}^I$ and $HT(\cdot, s) : \mathbb{R}^d \to \mathbb{R}^d$ be the hard thresholding operator, which keeps the largest $s$ entries (in magnitude) and sets the other entries equal to zero. Then for any $w^* \in \mathbb{R}^I$ such that $\|w^*\|_0 \leq s^*$, we have*

$$\|HT(w, s) - w\|^2 \leq \frac{|I| - s}{|I| - s^*}\|w - w^*\|^2.$$

***Lemma 2 ((Li et al., 2016b))** Let $w^* \in \mathbb{R}^d$ be the optimal sparse vector such that $\|w^*\|_0 \leq s^*$ and $HT(\cdot, s) : \mathbb{R}^d \to \mathbb{R}^d$ be the hard thresholding operator, which keeps the largest $s$ entries (in magnitude) and sets the other entries equal to zero. Given $s > s^*$ for any vector $w \in \mathbb{R}^d$, we have*

$$\|HT(w, s) - w^*\|^2 \leq (1 + \frac{2\sqrt{s^*}}{\sqrt{s - s^*}})\|w - w^*\|^2.$$

**Lemma 3 ((Nesterov, 2004))** *For any given vector $w, w' \in \mathbb{R}^d$, suppose $F(\cdot)$ also satisfies RSC and RSS conditions, then the following inequality holds*

$$\|\nabla F(w) - \nabla F(w')\|^2 \leq 2\rho_{\hat{s}}^+[F(w) - F(w') + \langle \nabla F(w'), w' - w \rangle].$$

**Lemma 4** *Suppose $F(w)$ is $\rho_{\hat{s}}^-$-strongly convex and each function $f_i(w)$ is $\rho_{\hat{s}}^+$-strongly smooth with parameter $\hat{s} = 2s + s^*$. $\mathcal{I} = \mathrm{supp}(w^*) \cup \mathrm{supp}(\widetilde{w}^r) \cup \mathrm{supp}(\widetilde{w}^{r+1})$, $|\mathcal{B}|$ is the batch size. For any $w^t \in \mathbb{R}^d$ and the sample $i_t \in \mathcal{B}$, denote $\nu_{i_t} = (\nabla_{\mathcal{I}} f_{i_t}(w^t) - \nabla_{\mathcal{I}} f_{i_t}(w^*)) - (\nabla_{\mathcal{I}} F(w^t) - \nabla_{\mathcal{I}} F(w^*))$, we have*

$$\mathbb{E}\left\|\frac{1}{|\mathcal{B}|}\sum_{i_t \in \mathcal{B}} \nu_{i_t}\right\|^2 = \frac{n - |\mathcal{B}|}{|\mathcal{B}|(n-1)}\mathbb{E}\|\nabla_{\mathcal{I}} f_i(w^t) - \nabla_{\mathcal{I}} f_i(w^*) + \nabla_{\mathcal{I}} F(w^*) - \nabla_{\mathcal{I}} F(w^t)\|^2. \quad (7)$$

*Proof.      See Section B.1 for the proof of Lemma 4.*                                                    □

**Lemma 5** *Suppose $F(w)$ is $\rho_{\hat{s}}^-$-strongly convex and each function $f_i(w)$ is $\rho_{\hat{s}}^+$-strongly smooth with parameter $\hat{s} = 2s + s^*$. Let $w^* \in \mathbb{R}^d$ be the optimal sparse vector with $\|w^*\|_0 \leq s^*$. $\mathcal{I} = \mathrm{supp}(w^*) \cup \mathrm{supp}(\widetilde{w}^r) \cup \mathrm{supp}(\widetilde{w}^{r+1})$, $\delta > 1$ is a uniform constant factor, and $|\mathcal{B}|$ is the batch size. For any $w^t, \widetilde{w} \in \mathbb{R}^d$ and the sample $i_t$, denote $\nabla_{\mathcal{S}} g(w^t) = \frac{1}{|\mathcal{B}|}\sum_{i_t \in \mathcal{B}} \nabla_{\mathcal{S}} f_{i_t}(w^t) - \nabla_{\mathcal{S}} f_{i_t}(\widetilde{w}) + \nabla_{\mathcal{S}} F(\widetilde{w})$, then we can bound $\mathbb{E}\|\nabla_{\mathcal{S}} g_{\mathcal{I}}(w^t)\|^2$ as follows:*

$$\mathbb{E}\|\nabla_{\mathcal{S}} g_{\mathcal{I}}(w^t)\|^2 \leq \frac{1}{k}(16\delta\rho_{\hat{s}}^+(1 + \frac{n - |\mathcal{B}|}{|\mathcal{B}|(n-1)})\mathbb{E}[F(w^t) - F(w^*)] + 4\|\nabla_{\mathcal{I}} F(w^*)\|^2$$
$$+ 16\delta\rho_{\hat{s}}^+ \frac{n - |\mathcal{B}|}{|\mathcal{B}|(n-1)}\mathbb{E}[F(\widetilde{w}) - F(w^*)]),$$

*Proof.      See Section B.2 for the detailed proof of Lemma 5.*                                          □

**Lemma 6** *Suppose $F(w)$ is $\rho_{\hat{s}}^-$-strongly convex and each function $f_i(w)$ is $\rho_{\hat{s}}^+$-strongly smooth with parameter $\hat{s} = 2s + s^*$. Let $w^* \in \mathbb{R}^d$ be the optimal sparse vector with $\|w^*\|_0 \leq s^*$. $\mathcal{I} = \mathrm{supp}(w^*) \cup \mathrm{supp}(\widetilde{w}^r) \cup \mathrm{supp}(\widetilde{w}^{r+1})$, $\delta > 1$ is a uniform constant factor, and $D_m = \max_{j=1,\cdots,d} \frac{1}{p_j}$. For any $w^t, \widetilde{w} \in \mathbb{R}^d$ and the sample $i_t$, denote $\nabla_{\mathcal{S}} g(w^t) = \nabla_{\mathcal{S}} f_{i_t}(w^t) - \nabla_{\mathcal{S}} f_{i_t}(\widetilde{w}) + D_{i_t}\nabla_{\mathcal{S}} F(\widetilde{w})$, we can bound $\mathbb{E}\|\nabla_{\mathcal{S}} g_{\mathcal{I}}(w^t)\|^2$ as follows:*

$$\mathbb{E}\|\nabla_{\mathcal{S}} g_{\mathcal{I}}(w^t)\|^2 \leq \frac{1}{k}(12\delta\rho_{\hat{s}}^+[F(w^t) - F(w^*)] + (3 + 12(D_m^2 - 1))\|\nabla_{\mathcal{I}} F(w^*)\|^2$$
$$+ (24\delta\rho_{\hat{s}}^+ + 48\delta\rho_{\hat{s}}^+(D_m^2 - 1))[F(\widetilde{w}) - F(w^*)]),$$

*where $p_j$ is the probability that dimension $j$ belonging to the support set of randomly sampling sample $i_t$.*

*Proof.*    See Section B.3 for the proof of Lemma 6.                                                    □

This lemma is different from other variance reduced results for sparsity-constrained problem (Mania et al., 2017; Johnson & Zhang, 2013; Chen & Gu, 2016; Shang et al., 2018). Since this variance reduced approximate gradients introduce an approximate full gradient to make fully sparse update in cardinality constraint problem.

# B    PROOFS OF KEY LEMMAS

In this section, we prove three key lemmas.

## B.1    PROOF OF LEMMA 4

*Proof.*    Let $\nu_{i_t} = (\nabla_{\mathcal{I}} f_{i_t}(w^t) - \nabla_{\mathcal{I}} f_{i_t}(w^*)) - (\nabla_{\mathcal{I}} F(w^t) - \nabla_{\mathcal{I}} F(w^*))$. We have

$$
\begin{aligned}
\mathbb{E}\left\| \frac{1}{|\mathcal{B}|} \sum_{i_t \in \mathcal{B}} \nu_{i_t} \right\|^2 &= \frac{1}{|\mathcal{B}|^2} \mathbb{E} \sum_{i_t, i'_t \in \mathcal{B}} \nu_{i_t}^\top \nu_{i'_t} \\
&= \frac{1}{|\mathcal{B}|^2} \mathbb{E} \sum_{i_t \neq i'_t \in \mathcal{B}} \nu_{i_t}^\top \nu_{i'_t} + \frac{1}{|\mathcal{B}|} \mathbb{E}\|\nu_i\|^2 \\
&= \frac{|\mathcal{B}| - 1}{|\mathcal{B}| n(n-1)} \sum_{i \neq i'} \nu_i^\top \nu_{i'} + \frac{1}{|\mathcal{B}|} \mathbb{E}\|\nu_i\|^2 \\
&= \frac{|\mathcal{B}| - 1}{|\mathcal{B}| n(n-1)} \sum_{i, i'} \nu_i^\top \nu_{i'} - \frac{|\mathcal{B}| - 1}{|\mathcal{B}|(n-1)} \mathbb{E}\|\nu_i\|^2 + \frac{1}{|\mathcal{B}|} \mathbb{E}\|\nu_i\|^2 \\
&= \frac{n - |\mathcal{B}|}{|\mathcal{B}|(n-1)} \mathbb{E}\|\nu_i\|^2.
\end{aligned}
\tag{11}
$$

This proof is similar to that of (Zheng & Kwok, 2016).                                                    □

## B.2    PROOF OF LEMMA 5

*Proof.*    It is obvious that the stochastic variance reduced gradient satisfies

$$
\mathbb{E}[\nabla g(w^t)] = \mathbb{E}\left[ \frac{1}{|\mathcal{B}|} \sum_{i_t \in \mathcal{B}} (\nabla f_{i_t}(w^t) - \nabla f_{i_t}(\widetilde{w})) + \mu \right] = \nabla F(w^t).
\tag{12}
$$

Thus, $\nabla g(w^t)$ is an unbiased estimator of the full gradient $\nabla F(w^t)$. Now, we bound $\mathbb{E}\|\nabla g_{\mathcal{I}}(w^t)\|^2$. For any $i \in \{1, 2, \cdots, n\}$, we define the following function

$$
h_i(w) = f_i(w) - f_i(w^*) - \langle \nabla f_i(w^*), w - w^* \rangle.
\tag{13}
$$

It is easy to find that $\nabla h_i(w^*) = 0$, which implies that $h_i(w^*) = \min_w h_i(w)$. For any vector $w$, we have

$$
\begin{aligned}
0 = h_i(w^*) &\leq \min_\eta h_i(w - \eta \nabla_{\mathcal{I}} h_i(w)) \\
&\leq \min_\eta h_i(w) - \eta \langle \nabla h_i(w), \nabla_{\mathcal{I}} h_i(w) \rangle + \frac{\eta^2 \rho_{\hat{s}}^+}{2} \|\nabla_{\mathcal{I}} h_i(w)\|^2 \\
&= \min_\eta h_i(w) - \eta \|\nabla_{\mathcal{I}} h_i(w)\|^2 + \frac{\eta^2 \rho_{\hat{s}}^+}{2} \|\nabla_{\mathcal{I}} h_i(w)\|^2 \\
&= h_i(w) - \frac{1}{2\rho_{\hat{s}}^+} \|\nabla_{\mathcal{I}} h_i(w)\|^2,
\end{aligned}
\tag{14}
$$

where the second inequality follows from the RSS condition, and the second equality holds due to the fact that $\langle \nabla h_i(w), \nabla_{\mathcal{I}} h_i(w) \rangle = \|\nabla_{\mathcal{I}} h_i(w)\|^2$, and the last equality holds due to the fact that $\eta = \frac{1}{\rho_{\hat{s}}^+}$ minimizes the function. Then, we have

$$
\|\nabla_{\mathcal{I}} f_i(w) - \nabla_{\mathcal{I}} f_i(w^*)\|^2 \leq 2\rho_{\hat{s}}^+ [f_i(w) - f_i(w^*) - \langle \nabla_{\mathcal{I}} f_i(w^*), w - w^* \rangle].
\tag{15}
$$

Since the sampling $i$ is chosen uniformly from $\{1, 2, \cdots, n\}$, we have

$$
\begin{aligned}
&\mathbb{E}\|\nabla_{\mathcal{I}} f_i(w) - \nabla_{\mathcal{I}} f_i(w^*)\|^2 \\
&= \mathbb{E}\|\nabla_{\mathcal{I}} F(w) - \nabla_{\mathcal{I}} F(w^*)\|^2 \\
&= \frac{1}{n} \sum_{i=1}^n \|\nabla_{\mathcal{I}} f_i(w) - \nabla_{\mathcal{I}} f_i(w^*)\|^2 \\
&\leq 2\rho_{\widehat{s}}^+ [F(w) - \mathcal{F}(w^*) - \langle \nabla_{\mathcal{I}} F(w^*), w - w^* \rangle] \\
&\leq 2\rho_{\widehat{s}}^+ [F(w) - F(w^*) + |\langle \nabla_{\mathcal{I}} F(w^*), w - w^* \rangle|] \\
&\leq 4\delta\rho_{\widehat{s}}^+ [F(w) - F(w^*)],
\end{aligned}
\tag{16}
$$

where the last inequality follows from the restricted strong convexity of $F(w)$. And we use a simple underlying consensus that there must exist a uniform factor $\delta > 1$ which makes the following inequality true:

$$
\delta(F(w) - F(w^*)) \geq |\langle \nabla_{\mathcal{I}} F(w^*), w - w^* \rangle| + \frac{\rho_{\widehat{s}}^-}{2}\|w - w^*\|^2.
$$

This property is called the extended restricted strong convexity used in the following proof. Therefore, we have

$$
\begin{aligned}
\mathbb{E}\|\nabla g_{\mathcal{I}}(w^t)\|^2 &= \mathbb{E}\left\| \frac{1}{|\mathcal{B}|} \sum_{i_t \in \mathcal{B}} \left( \nabla_{\mathcal{I}} f_{i_t}(w^t) - \nabla_{\mathcal{I}} f_{i_t}(\widetilde{w}) \right) + \nabla_{\mathcal{I}} F(\widetilde{w}) \right\|^2 \\
&= \mathbb{E}\left\| \left( \frac{1}{|\mathcal{B}|} \sum_{i_t \in \mathcal{B}} \nabla_{\mathcal{I}} f_{i_t}(w^t) - \nabla_{\mathcal{I}} F(w^t) - \nabla_{\mathcal{I}} f_{i_t}(w^*) + \nabla_{\mathcal{I}} F(w^*) \right) \right. \\
&\quad + \nabla_{\mathcal{I}} F(w^t) - \nabla_{\mathcal{I}} F(w^*) + \nabla_{\mathcal{I}} F(w^*) \\
&\quad \left. - \left( \frac{1}{|\mathcal{B}|} \sum_{i_t \in \mathcal{B}} \nabla_{\mathcal{I}} f_{i_t}(\widetilde{w}) - \nabla_{\mathcal{I}} F(\widetilde{w}) - \nabla_{\mathcal{I}} f_{i_t}(w^*) + \nabla_{\mathcal{I}} F(w^*) \right) \right\|^2 \\
&\leq 4\mathbb{E}\left\| \frac{1}{|\mathcal{B}|} \sum_{i_t \in \mathcal{B}} \nabla_{\mathcal{I}} f_{i_t}(w^t) - \nabla_{\mathcal{I}} F(w^t) - \nabla_{\mathcal{I}} f_{i_t}(w^*) + \nabla_{\mathcal{I}} F(w^*) \right\|^2 \\
&\quad + 4\mathbb{E}\left\| \frac{1}{|\mathcal{B}|} \sum_{i_t \in \mathcal{B}} \nabla_{\mathcal{I}} f_{i_t}(\widetilde{w}) - \nabla_{\mathcal{I}} F(\widetilde{w}) - \nabla_{\mathcal{I}} f_{i_t}(w^*) + \nabla_{\mathcal{I}} F(w^*) \right\|^2 \\
&\quad + 4\mathbb{E}\|\nabla_{\mathcal{I}} F(w^t) - \nabla_{\mathcal{I}} F(w^*)\|^2 + 4\|\nabla_{\mathcal{I}} F(w^*)\|^2 \\
&= 4\frac{n - |\mathcal{B}|}{|\mathcal{B}|(n-1)} \mathbb{E}\|\nabla_{\mathcal{I}} f_i(w^t) - \nabla_{\mathcal{I}} F(w^t) - \nabla_{\mathcal{I}} f_i(w^*) + \nabla_{\mathcal{I}} F(w^*)\|^2 \\
&\quad + 4\frac{n - |\mathcal{B}|}{|\mathcal{B}|(n-1)} \mathbb{E}\|\nabla_{\mathcal{I}} f_i(\widetilde{w}) - \nabla_{\mathcal{I}} F(\widetilde{w}) - \nabla_{\mathcal{I}} f_i(w^*) + \nabla_{\mathcal{I}} F(w^*)\|^2 \\
&\quad + 4\mathbb{E}\|\nabla_{\mathcal{I}} F(w^t) - \nabla_{\mathcal{I}} F(w^*)\|^2 + 4\|\nabla_{\mathcal{I}} F(w^*)\|^2 \\
&\leq \left(4 + 4\frac{n - |\mathcal{B}|}{|\mathcal{B}|(n-1)}\right) \mathbb{E}\|\nabla_{\mathcal{I}} f_i(w^t) - \nabla_{\mathcal{I}} f_i(w^*)\|^2 + 4\|\nabla_{\mathcal{I}} F(w^*)\|^2 \\
&\quad + 4\frac{n - |\mathcal{B}|}{|\mathcal{B}|(n-1)} \mathbb{E}\|\nabla_{\mathcal{I}} f_i(\widetilde{w}) - \nabla_{\mathcal{I}} f_i(w^*)\|^2 \\
&\leq 16\delta\rho_{\widehat{s}}^+ \left(1 + \frac{n - |\mathcal{B}|}{|\mathcal{B}|(n-1)}\right) \mathbb{E}[F(w^t) - F(w^*)] + 4\|\nabla_{\mathcal{I}} F(w^*)\|^2 \\
&\quad + 16\delta\rho_{\widehat{s}}^+ \frac{n - |\mathcal{B}|}{|\mathcal{B}|(n-1)} \mathbb{E}[F(\widetilde{w}) - F(w^*)]
\end{aligned}
\tag{17}
$$

where the first inequality follows from $\|a + b + c + d\|^2 \leq 4\|a\|^2 + 4\|b\|^2 + 4\|c\|^2 + 4\|d\|^2$, the third equality follows from Lemma 4, the second inequality follows from $\mathbb{E}\|x - \mathbb{E}x\|^2 \leq \mathbb{E}\|x\|^2$ and (16),

and the last inequality follows from (16). Due to the fact that $\mathbb{E}\|\nabla_{\mathcal{S}} g_{\mathcal{I}}(w^t)\|^2 \leq 1/k \mathbb{E}\|\nabla g_{\mathcal{I}}(w^t)\|^2$ where $\mathcal{I} = supp(w^*) \cup supp(\widetilde{w}^{r+1}) \cup supp(\widetilde{w}^r)$ and $\mathcal{S} = \mathcal{G}_{j_t} \cup supp(\widetilde{w}^r)$, we have

$$
\begin{aligned}
\mathbb{E}\|\nabla_{\mathcal{S}} g_{\mathcal{I}}(w^t)\|^2 \leq &\frac{1}{k}(16\delta\rho_{\hat{s}}^+(1 + \frac{n - |\mathcal{B}|}{|\mathcal{B}|(n-1)})\mathbb{E}[F(w^t) - F(w^*)] + 4\|\nabla_{\mathcal{I}} F(w^*)\|^2 \\
&+ 16\delta\rho_{\hat{s}}^+ \frac{n - |\mathcal{B}|}{|\mathcal{B}|(n-1)}\mathbb{E}[F(\widetilde{w}) - F(w^*)]),
\end{aligned}
\tag{18}
$$

where $k$ is the number of blocks. Note that we take expectation with respect to randomized block in the above equation. This completes the proof. $\qquad\square$

### B.3 PROOF OF LEMMA 6

*Proof.* By the definition $\nabla g(w^t) = \nabla f_{i_t}(w^t) - \nabla f_{i_t}(\widetilde{w}) + D_{i_t}\nabla F(\widetilde{w})$, we have $\mathbb{E}[D_{i_t}\nabla F(\widetilde{w})] = \nabla F(\widetilde{w})$. Therefore, $\nabla g(w^t)$ is an unbiased estimator of $\nabla F(w^t)$.

Based on (16), we have the following result:

$$
\begin{aligned}
\mathbb{E}\|\nabla g_{\mathcal{I}}(w^t)\|^2 = &\mathbb{E}\|\nabla_{\mathcal{I}} f_{i_t}(w^t) - \nabla_{\mathcal{I}} f_{i_j}(\widetilde{w}) + D_{i_t}\nabla_{\mathcal{I}} F(\widetilde{w})\|^2 \\
\leq &3\mathbb{E}\|[\nabla_{\mathcal{I}} f_{i_t}(\widetilde{w}) - \nabla_{\mathcal{I}} f_{i_t}(w^*)] - \nabla_{\mathcal{I}} F(\widetilde{w}) + \nabla_{\mathcal{I}} F(w^*) \\
&+ \nabla_{\mathcal{I}} F(\widetilde{w}) - D_{i_t}\nabla_{\mathcal{I}} F(\widetilde{w})\|^2 \\
&+ 3\mathbb{E}\|\nabla_{\mathcal{I}} f_{i_t}(w^t) - \nabla_{\mathcal{I}} f_{i_t}(w^*)\|^2 + 3\|\nabla_{\mathcal{I}} F(w^*)\|^2 \\
\leq &6\mathbb{E}\|[\nabla_{\mathcal{I}} f_{i_t}(\widetilde{w}) - \nabla_{\mathcal{I}} f_{i_t}(w^*)] - \nabla_{\mathcal{I}} F(\widetilde{w}) + \nabla_{\mathcal{I}} F(w^*)\|^2 \\
&+ 6\mathbb{E}\|\nabla_{\mathcal{I}} F(\widetilde{w}) - D\nabla_{\mathcal{I}} F(\widetilde{w})\|^2 \\
&+ 3\mathbb{E}\|\nabla_{\mathcal{I}} f_{i_t}(w^t) - \nabla_{\mathcal{I}} f_{i_t}(w^*)\|^2 + 3\|\nabla_{\mathcal{I}} F(w^*)\|^2 \\
\leq &12\delta\rho_{\hat{s}}^+[F(w^t) - F(w^*)] + (3 + 12(D_m^2 - 1))\|\nabla_{\mathcal{I}} F(w^*)\|^2 \\
&+ (24\delta\rho_{\hat{s}}^+ + 48\delta\rho_{\hat{s}}^+(D_m^2 - 1))[F(\widetilde{w}) - F(w^*)],
\end{aligned}
\tag{19}
$$

where the first inequality follows from $\|a + b + c\|^2 \leq 3\|a\|^2 + 3\|b\|^2 + 3\|c\|^2$, the second inequality follows from $\|a + b\|^2 \leq 2\|a\|^2 + 2\|b\|^2$, the third inequality follows from $\mathbb{E}\|w - \mathbb{E}w\|^2 \leq \mathbb{E}\|w\|^2$ with $\mathbb{E}[\nabla_{\mathcal{I}} f_{i_t}(\widetilde{w}) - \nabla_{\mathcal{I}} f_{i_t}(w^*)] = \nabla_{\mathcal{I}} F(\widetilde{w}) - \nabla_{\mathcal{I}} F(w^*)$, and $\mathbb{E}\|\nabla_{\mathcal{I}} f_i(w^t) - \nabla_{\mathcal{I}} f_i(w^*)\|^2 \leq 4\delta\rho_{\hat{s}}^+[F(w^t) - F(w^*)]$ with $\delta > 1$, which is also satisfied for $\widetilde{w}$ with $\delta > 1$. Since $D_m = \max_{v=1,\cdots,d} \frac{1}{p_v}$, we bound the $\|\nabla_{\mathcal{I}} F(\widetilde{w}) - D\nabla_{\mathcal{I}} F(\widetilde{w})\|^2$ term as follows:

$$
\begin{aligned}
&\|(D - I)\nabla_{\mathcal{I}} F(\widetilde{w})\|^2 \\
=&\sum_{v=1}^{d}(\frac{1}{p_v^2} - 1)[\nabla_{\mathcal{I}} F(\widetilde{w})]_v^2 \\
\leq&(D_m^2 - 1)\|\nabla_{\mathcal{I}} F(\widetilde{w})\|^2 \\
=&(D_m^2 - 1)\|\nabla_{\mathcal{I}} F(\widetilde{w}) - \nabla_{\mathcal{I}} F(w^*) + \nabla_{\mathcal{I}} F(w^*)\|^2 \\
\leq&2(D_m^2 - 1)\|\nabla_{\mathcal{I}} F(\widetilde{w}) - \nabla_{\mathcal{I}} F(w^*)\|^2 + 2(D_m^2 - 1)\|\nabla_{\mathcal{I}} F(w^*)\|^2 \\
\leq&8\delta\rho_{\hat{s}}^+(D_m^2 - 1)[F(\widetilde{w}) - F(w^*)] + 2(D_m^2 - 1)\|\nabla_{\mathcal{I}} F(w^*)\|^2.
\end{aligned}
\tag{20}
$$

Due to the fact that $\mathbb{E}\|\nabla_{\mathcal{S}} g_{\mathcal{I}}(w^t)\|^2 \leq \frac{1}{k}\mathbb{E}\|\nabla g_{\mathcal{I}}(w^t)\|^2$, we have

$$
\begin{aligned}
\mathbb{E}\|\nabla_{\mathcal{S}} g_{\mathcal{I}}(w^t)\|^2 \leq &\frac{1}{k}(12\delta\rho_{\hat{s}}^+[F(w^t) - F(w^*)] + (3 + 12(D_m^2 - 1))\|\nabla_{\mathcal{I}} F(w^*)\|^2 \\
&+ (24\delta\rho_{\hat{s}}^+ + 48\delta\rho_{\hat{s}}^+(D_m^2 - 1))[F(\widetilde{w}) - F(w^*)]),
\end{aligned}
\tag{21}
$$

where $k$ is the number of blocks. Note that we take expectation with respect to randomized block in the above equation. This completes the proof. $\qquad\square$

## C PROOFS OF THEOREM 1 AND COROLLARIES 1 AND 2

In this section, we give the detailed proofs of Theorem 1 and Corollaries 1 and 2, which can guarantee the convergence properties of our SBCD-HTP.

## C.1 PROOF OF THEOREM 1

*Proof.* In this subsection, we provide the proof of Theorem 1. Let $w^{t+1} = w^t - \eta\nabla_{\mathcal{S}}g_{\mathcal{I}}(w^t)$ and $\mathcal{I} = \mathcal{I}^* \cup \mathcal{I}^r \cup \mathcal{I}^{r+1}$, where $\mathcal{I}^* = \text{supp}(w^*)$, $\mathcal{I}^r = \text{supp}(\widetilde{w}^r)$ and $\mathcal{I}^{r+1} = \text{supp}(\widetilde{w}^{r+1})$. Conditioning on $w^t$, we have the following expectation

$$\mathbb{E}\|w^{t+1} - w^*\|^2$$

$$= \mathbb{E}\|w^t - \eta\nabla_{\mathcal{S}}g_{\mathcal{I}}(w^t) - w^*\|^2$$

$$= \mathbb{E}\|w^t - w^*\|^2 + \eta^2\mathbb{E}\|\nabla_{\mathcal{S}}g_{\mathcal{I}}(w^t)\|^2 - 2\eta\langle w^t - w^*, \mathbb{E}\nabla_{\mathcal{S}}g_{\mathcal{I}}(w^t)\rangle$$

$$\leq \mathbb{E}\|w^t - w^*\|^2 + \eta^2\mathbb{E}\|\nabla_{\mathcal{S}}g_{\mathcal{I}}(w^t)\|^2 + \frac{2\eta}{k}\langle w^* - w^t, \nabla_{\mathcal{I}}F(w^t)\rangle$$

$$\leq \mathbb{E}\|w^t - w^*\|^2 + \eta^2\mathbb{E}\|\nabla_{\mathcal{S}}g_{\mathcal{I}}(w^t)\|^2 + \frac{2\eta\delta}{k}[F(w^*) - F(w^t)]$$

$$\leq \mathbb{E}\|w^t - w^*\|^2 + \frac{1}{k}(16\delta\rho_{\hat{s}}^+\eta^2(1 + \frac{n-|\mathcal{B}|}{|\mathcal{B}|(n-1)})\mathbb{E}[F(w^t) - F(w^*)] \tag{22}$$

$$+ 16\delta\rho_{\hat{s}}^+\eta^2\frac{n-|\mathcal{B}|}{|\mathcal{B}|(n-1)}\mathbb{E}[F(\widetilde{w}) - F(w^*)] - 2\eta\delta[F(w^t) - F(w^*)]) + \frac{4\eta^2}{k}\|\nabla_{\mathcal{I}}F(w^*)\|^2$$

$$= \mathbb{E}\|w^t - w^*\|^2 + \frac{1}{k}((16\delta\rho_{\hat{s}}^+\eta^2(1 + \frac{n-|\mathcal{B}|}{|\mathcal{B}|(n-1)}) - 2\eta\delta)\mathbb{E}[F(w^t) - F(w^*)]$$

$$+ 16\delta\rho_{\hat{s}}^+\eta^2\frac{n-|\mathcal{B}|}{|\mathcal{B}|(n-1)}\mathbb{E}[F(\widetilde{w}) - F(w^*)]) + \frac{4\eta^2}{k}\|\nabla_{\mathcal{I}}F(w^*)\|^2$$

where the first inequality follows from the expectation with respect to randomized block, the second inequality follows from our extended restricted strong convexity in the proof of Lemma 5, and the third inequality holds by Lemma 5. Notice that $\widetilde{w} = w^0 = \widetilde{w}^{r-1}$. Summing (22) over $t = 0, \cdots, m-1$ and taking expectation with respect to all randomness, we have

$$\mathbb{E}\|w^m - w^*\|^2 \leq \mathbb{E}\|\widetilde{w}^{r-1} - w^*\|^2 + \frac{1}{k}((16\delta\rho_{\hat{s}}^+\eta^2m(1 + \frac{n-|\mathcal{B}|}{|\mathcal{B}|(n-1)}) - 2m\eta\delta)\mathbb{E}[F(w^m) - F(w^*)]$$

$$+ 16\delta\rho_{\hat{s}}^+\eta^2m\frac{n-|\mathcal{B}|}{|\mathcal{B}|(n-1)}\mathbb{E}[F(\widetilde{w}^{r-1}) - F(w^*)]) + \frac{4\eta^2m}{k}\|\nabla_{\mathcal{I}}F(w^*)\|^2 \tag{23}$$

Notice that the convergence property used here is similar to the proofs of the convergence results for many hard thresholding algorithms (Chen & Gu, 2016; 2017; Li et al., 2016b).

Moreover, we can obtain

$$\mathbb{E}\|w^m - w^*\|^2 \leq \mathbb{E}\|\widetilde{w}^{r-1} - w^*\|^2 + \frac{1}{k}((16\delta\rho_{\hat{s}}^+\eta^2m(1 + \frac{n-|\mathcal{B}|}{|\mathcal{B}|(n-1)}) - 2m\eta\delta)\mathbb{E}[F(w^m) - F(w^*)]$$

$$+ 16\delta\rho_{\hat{s}}^+\eta^2m\frac{n-|\mathcal{B}|}{|\mathcal{B}|(n-1)}\mathbb{E}[F(\widetilde{w}^{r-1}) - F(w^*)]) + \frac{4\eta^2m}{k}\|\nabla_{\mathcal{I}}F(w^*)\|^2$$

$$= \mathbb{E}\|\widetilde{w}^{r-1} - w^*\|^2 + \frac{1}{k}((16\delta\rho_{\hat{s}}^+\eta^2m(1 + \frac{n-|\mathcal{B}|}{|\mathcal{B}|(n-1)}) - 2m\eta\delta)\mathbb{E}[F(\widetilde{w}^r) - F(w^*)]$$

$$- (16\delta\rho_{\hat{s}}^+\eta^2m(1 + \frac{n-|\mathcal{B}|}{|\mathcal{B}|(n-1)}) - 2m\eta\delta)\mathbb{E}[F(\widetilde{w}^r) - F(w^m)]$$

$$+ 16\delta\rho_{\hat{s}}^+\eta^2m\frac{n-|\mathcal{B}|}{|\mathcal{B}|(n-1)}\mathbb{E}[F(\widetilde{w}^{r-1}) - F(w^*)]) + \frac{4\eta^2m}{k}\|\nabla_{\mathcal{I}}F(w^*)\|^2$$

$$\leq \mathbb{E}\|\widetilde{w}^{r-1} - w^*\|^2 + \frac{1}{k}((16\delta\rho_{\hat{s}}^+\eta^2m(1 + \frac{n-|\mathcal{B}|}{|\mathcal{B}|(n-1)}) - 2m\eta\delta)\mathbb{E}[F(\widetilde{w}^r) - F(w^*)]$$

$$- (16\delta\rho_{\hat{s}}^+\eta^2m(1 + \frac{n-|\mathcal{B}|}{|\mathcal{B}|(n-1)}) - 2m\eta\delta)\frac{\rho_s^+(d-s)}{2\sigma(d-s^*)}\mathbb{E}\|w^m - w^*\|^2$$

$$+ 16\delta\rho_{\hat{s}}^+\eta^2m\frac{n-|\mathcal{B}|}{|\mathcal{B}|(n-1)}\mathbb{E}[F(\widetilde{w}^{r-1}) - F(w^*)]) + \frac{4\eta^2m}{k}\|\nabla_{\mathcal{I}}F(w^*)\|^2 \tag{24}$$

where the second inequality holds due to the fact that we have the following results: When $F(\widetilde{w}^r) - F(w^m) > 0$, by using Lemma 3 on $w^m$ and $\widetilde{w}^r$, we have

$$
\begin{aligned}
\langle \nabla F(w^m), \widetilde{w}^r - w^m \rangle &\leq F(\widetilde{w}^r) - F(w^m) - \frac{1}{2\rho_{\widehat{s}}^+} \|\nabla F(\widetilde{w}^r) - \nabla F(w^m)\|^2 \\
&\leq (1-\sigma)[F(\widetilde{w}^r) - F(w^m)],
\end{aligned}
\tag{25}
$$

where the last inequality holds due to the fact that we assume that there exists a constant factor $\sigma > 0$ making this inequality true. Then by using the RSS condition, we have

$$
\begin{aligned}
F(\widetilde{w}^r) - F(w^m) &\leq \langle \nabla F(w^m), \widetilde{w}^r - w^m \rangle + \frac{\rho_{\widehat{s}}^+}{2} \|\widetilde{w}^r - w^m\|^2 \\
&\leq (1-\sigma)[F(\widetilde{w}^r) - F(w^m)] + \frac{\rho_{\widehat{s}}^+}{2} \|\widetilde{w}^r - w^m\|^2.
\end{aligned}
\tag{26}
$$

After simplification, we have $F(\widetilde{w}^r) - F(w^m) \leq \frac{\rho_{\widehat{s}}^+}{2\sigma} \|\widetilde{w}^r - w^m\|^2$. Using Lemma 1, we have

$$
\|\widetilde{w}^r - w^m\|^2 = \|\mathcal{H}_k(w^m) - w^m\|^2 \leq \frac{|I| - s}{|I| - s^*} \|w^m - w^*\|^2,
\tag{27}
$$

and

$$
F(\widetilde{w}^r) - F(w^m) \leq \frac{\rho_{\widehat{s}}^+ (d - s)}{2\sigma(d - s^*)} \|w^m - w^*\|^2.
\tag{28}
$$

When $F(\widetilde{w}^r) - F(w^m) < 0$, we can omit $-(16\delta\rho_{\widehat{s}}^+ \eta^2 m(1 + \frac{n-|\mathcal{B}|}{|\mathcal{B}|(n-1)}) - 2m\eta\delta)\mathbb{E}[F(\widetilde{w}^r) - F(w^m)]$ directly since $16\delta\rho_{\widehat{s}}^+ \eta^2 m(1 + \frac{n-|\mathcal{B}|}{|\mathcal{B}|(n-1)}) - 2m\eta\delta$ is less than zero. Because this is the simple situation, we only analyze the complex one in our whole proof. Therefore, we have the following inequality:

$$
\begin{aligned}
&(1 + \frac{1}{k}(16\delta\rho_{\widehat{s}}^+ \eta^2 m(1 + \frac{n-|\mathcal{B}|}{|\mathcal{B}|(n-1)}) - 2m\eta\delta)\frac{\rho_{\widehat{s}}^+(d-s)}{2\sigma(d-s^*)})\mathbb{E}\|w^m - w^*\|^2 \\
&\leq \mathbb{E}\|\widetilde{w}^{r-1} - w^*\|^2 + \frac{1}{k}((16\delta\rho_{\widehat{s}}^+ \eta^2 m(1 + \frac{n-|\mathcal{B}|}{|\mathcal{B}|(n-1)}) - 2m\eta\delta)\mathbb{E}[F(\widetilde{w}^r) - F(w^*)] \\
&\quad + 16\delta\rho_{\widehat{s}}^+ \eta^2 m \frac{n-|\mathcal{B}|}{|\mathcal{B}|(n-1)} \mathbb{E}[F(\widetilde{w}^{r-1}) - F(w^*)]) + \frac{4\eta^2 m}{k} \|\nabla_{\mathcal{I}} F(w^*)\|^2
\end{aligned}
\tag{29}
$$

Since $\widetilde{w}^r = \mathcal{H}_k(w^m)$, i.e., $\widetilde{w}^r$ is the best $s$-sparse approximation of $w^m$, then we have the following result due to Lemma 2,

$$
\|\widetilde{w}^r - w^*\|^2 \leq (1 + \frac{2\sqrt{s^*}}{\sqrt{s - s^*}})\|w^m - w^*\|^2.
\tag{30}
$$

Let $\alpha = 1 + \frac{2\sqrt{s^*}}{\sqrt{s-s^*}}$, and by combining (29) and (30), we have

$$
\begin{aligned}
&(1 + \frac{1}{k}(16\delta\rho_{\widehat{s}}^+ \eta^2 m(1 + \frac{n-|\mathcal{B}|}{|\mathcal{B}|(n-1)}) - 2m\eta\delta)\frac{\rho_{\widehat{s}}^+(d-s)}{2\sigma(d-s^*)})\mathbb{E}\|\widetilde{w}^r - w^*\|^2 \\
&\leq \alpha\mathbb{E}\|\widetilde{w}^{r-1} - w^*\|^2 + \frac{\alpha}{k}((16\delta\rho_{\widehat{s}}^+ \eta^2 m(1 + \frac{n-|\mathcal{B}|}{|\mathcal{B}|(n-1)}) - 2m\eta\delta)\mathbb{E}[F(\widetilde{w}^r) - F(w^*)] \\
&\quad + 16\delta\rho_{\widehat{s}}^+ \eta^2 m \frac{n-|\mathcal{B}|}{|\mathcal{B}|(n-1)} \mathbb{E}[F(\widetilde{w}^{r-1}) - F(w^*)]) + \frac{4\eta^2 m\alpha}{k} \|\nabla_{\mathcal{I}} F(w^*)\|^2 \\
&\leq \frac{2\alpha}{\rho_{\widehat{s}}^-}\mathbb{E}[F(\widetilde{w}^{r-1}) - F(w^*)] + \frac{\alpha}{k}((16\delta\rho_{\widehat{s}}^+ \eta^2 m(1 + \frac{n-|\mathcal{B}|}{|\mathcal{B}|(n-1)}) - 2m\eta\delta)\mathbb{E}[F(\widetilde{w}^r) - F(w^*)] \\
&\quad + 16\delta\rho_{\widehat{s}}^+ \eta^2 m \frac{n-|\mathcal{B}|}{|\mathcal{B}|(n-1)} \mathbb{E}[F(\widetilde{w}^{r-1}) - F(w^*)]) + \frac{4\eta^2 m\alpha}{k} \|\nabla_{\tilde{\mathcal{I}}} F(w^*)\|^2
\end{aligned}
\tag{31}
$$

where the last inequality follows from the RSC condition and the definition of $\tilde{\mathcal{I}}$, i.e., $\tilde{\mathcal{I}} = \text{supp}(HT(\nabla F(w^*), 2s)) \cup \text{supp}(w^*)$.

Through proper simplification, we have the following result:

$$\frac{\alpha}{k}(2m\eta\delta - 16\delta\rho_{\widehat{s}}^+\eta^2 m(1 + \frac{n-|\mathcal{B}|}{|\mathcal{B}|(n-1)}))\mathbb{E}[F(\widetilde{w}^r) - F(w^*)]$$
$$\leq (\frac{2\alpha}{\rho_{\widehat{s}}^-} + 16\delta\rho_{\widehat{s}}^+\eta^2 m\alpha\frac{n-|\mathcal{B}|}{|\mathcal{B}|k(n-1)})\mathbb{E}[F(\widetilde{w}^{r-1}) - F(w^*)] + \frac{4\eta^2 m\alpha}{k}\|\nabla_{\widetilde{\mathcal{I}}}F(w^*)\|^2 \tag{32}$$

where the first term of the LHS of (31) is omitted due to the fact that the coefficient of $\mathbb{E}\|\widetilde{w}^r - w^*\|^2$ is larger than zero, when $8\rho_{\widehat{s}}^+ m\eta^2(1 + \frac{n-|\mathcal{B}|}{|\mathcal{B}|(n-1)}) - m\eta + \frac{\sigma k(d-s^*)}{\delta\rho_{\widehat{s}}^+(d-s)} > 0$, which can be naturally satisfied under our parameter setting with $s \geq \kappa_{\widehat{s}}s^*$. Following from (32), we have

$$\mathbb{E}[F(\widetilde{w}^r) - F(w^*)] \leq \beta\mathbb{E}[F(\widetilde{w}^{r-1}) - F(w^*)] + \frac{2\eta}{\delta(1-\omega)}\|\nabla_{\widetilde{\mathcal{I}}}F(w^*)\|^2 \tag{33}$$

where $\beta = \frac{k}{\eta m\rho_{\widehat{s}}^-\delta(1-\omega)} + \frac{8\rho_{\widehat{s}}^+\eta(n-|\mathcal{B}|)}{|\mathcal{B}|(1-\omega)(n-1)}$ and $\omega = 8\rho_{\widehat{s}}^+\eta(1+\frac{n-|\mathcal{B}|}{|\mathcal{B}|(n-1)})$. By applying (33) recursively, then we have the following desired bound with $\beta \leq \frac{1}{2} < 1$:

$$\mathbb{E}[F(\widetilde{w}^r) - F(w^*)] \leq (\frac{1}{2})^r\mathbb{E}[F(\widetilde{w}^0) - F(w^*)] + \frac{\eta}{1-\omega}\|\nabla_{\widetilde{\mathcal{I}}}F(w^*)\|^2. \tag{34}$$

This completes the proof. $\qquad\square$

### C.2 PROOF OF COROLLARY 1

*Proof.* We then give the proof for the statistical estimation analysis. Following from the RSC condition of $F(\cdot)$, we have

$$F(w^*) \leq F(\widetilde{w}^r) + \langle\nabla F(w^*), w^* - \widetilde{w}^r\rangle - \frac{\rho_{\widehat{s}}^-}{2}\|\widetilde{w}^r - w^*\|^2. \tag{35}$$

Let $\phi = (\frac{1}{2})^r\mathbb{E}[F(\widetilde{w}^0) - F(w^*)] + \frac{\eta}{1-\omega}\|\nabla_{\widetilde{\mathcal{I}}}F(w^*)\|^2$. Combining (34) and (35), we have

$$\mathbb{E}[F(\widetilde{w}^r) - \phi] \leq F(w^*) \leq F(\widetilde{w}^r) + \langle\nabla F(w^*), w^* - \widetilde{w}^r\rangle - \frac{\rho_{\widehat{s}}^-}{2}\|\widetilde{w}^r - w^*\|^2. \tag{36}$$

Using the duality of norms, we have

$$\mathbb{E}\langle\nabla F(w^*), w^* - \widetilde{w}^r\rangle \leq \|\nabla F(w^*)\|_\infty\mathbb{E}\|\widetilde{w}^r - w^*\|_1 \leq \sqrt{\widehat{s}}\|\nabla F(w^*)\|_\infty\mathbb{E}\|\widetilde{w}^r - w^*\|. \tag{37}$$

Combining (36), (37) and $\|\mathbb{E}w\|^2 \leq \mathbb{E}\|w\|^2$, we have

$$\frac{\rho_{\widehat{s}}^-}{2}(\mathbb{E}\|\widetilde{w}^r - w^*\|)^2 \leq \sqrt{\widehat{s}}\|\nabla F(w^*)\|_\infty\mathbb{E}\|\widetilde{w}^r - w^*\| + \phi. \tag{38}$$

Let $a = \mathbb{E}\|\widetilde{w}^r - w^*\|$. From the above inequality, we solve the following quadratic function with respect to $a$,

$$\frac{\rho_{\widehat{s}}^-}{2}a^2 - \sqrt{\widehat{s}}\|\nabla F(w^*)\|_\infty a - \phi \leq 0, \tag{39}$$

which yields the bound

$$\mathbb{E}\|\widetilde{w}^r - w^*\| \leq \sqrt{\frac{2(\frac{1}{2})^r[F(\widetilde{w}^0) - F(w^*)]}{\rho_{\widehat{s}}^-}} + \sqrt{\frac{2\eta}{1-\omega}}\|\nabla_{\widetilde{\mathcal{I}}}F(w^*)\| + \frac{2\sqrt{\widehat{s}}\|\nabla F(w^*)\|_\infty}{\rho_{\widehat{s}}^-}$$
$$\leq \sqrt{\frac{2(\frac{1}{2})^r[F(\widetilde{w}^0) - F(w^*)]}{\rho_{\widehat{s}}^-}} + \left(\frac{2}{\rho_{\widehat{s}}^-} + \frac{2\eta}{1-\omega}\right)\sqrt{\widehat{s}}\|\nabla\mathcal{F}(w^*)\|_\infty. \tag{40}$$

This completes the proof. $\qquad\square$

## C.3 BOUND $\beta < 1$

Now we show that we can guarantee that $\beta \leq 1$ by providing the appropriate choices of constants with $\eta$, $k$ and $m$ used in the theorem. More specifically, let $\eta \leq \frac{C_3}{\rho_{\widehat{s}}^+} \leq \frac{1}{48\rho_{\widehat{s}}^+}$, and $|\mathcal{B}| = 1$, then we have

$$\frac{8\rho_{\widehat{s}}^+ \eta(n - |\mathcal{B}|)}{|\mathcal{B}|(1 - \omega)(n - 1)} \leq \frac{8C_3(n - |\mathcal{B}|)}{|\mathcal{B}|(n - 1)(1 - 8C_3(1 + \frac{n - |\mathcal{B}|}{|\mathcal{B}|(n-1)}))} \leq \frac{1}{4}. \tag{41}$$

If $\eta \geq \frac{C_2}{\rho_{\widehat{s}}^+}$ with $C_2 \leq C_3$, then we have

$$\frac{k}{\eta\delta m \rho_{\widehat{s}}^-(1 - \omega)} \leq \frac{1}{\frac{2\delta m C_2}{3k\kappa_{\widehat{s}}}}. \tag{42}$$

If we guarantee $\frac{1}{\frac{4\delta m C_2}{5k\kappa_{\widehat{s}}}} \leq \frac{1}{4}$, then we have

$$m \geq \frac{6k\kappa_{\widehat{s}}}{\delta C_2} = C_4 \kappa_{\widehat{s}}, \tag{43}$$

where $\delta > 1$. If we choose $C_2 = \frac{1}{60}, C_3 = \frac{1}{48}, C_4 = 1800, k = 10$ and $\delta = 2$, then we have $\beta \leq \frac{1}{2}$. Note that the convergence rate $\beta$ does not include $\alpha$, therefore, the required value of $k$ comes from the restricted constraint of the coefficient of $\|\widetilde{w}^r - w^*\|^2$, which implys that $s = \Omega(\kappa_{\widehat{s}} s^*)$.

## C.4 PROOF OF COROLLARY 2

*Proof.*

$$\mathbb{E}[F(\widetilde{w}^r) - F(w^*)] \leq \epsilon + \frac{\eta}{1 - \omega}\|\nabla_{\widetilde{\mathcal{I}}} F(w^*)\|^2. \tag{44}$$

Let $\varrho_1, \varrho_2, \cdots$ be a non-negative sequence of random variables which is defined as

$$\varrho_r = \max\left\{F(\widetilde{w}^r) - F(w^*) - \frac{\eta}{1 - \omega}\|\nabla_{\widetilde{\mathcal{I}}} F(w^*)\|^2, 0\right\}. \tag{45}$$

For a fixed $\epsilon > 0$, it follows from (34) and the Markov inequality,

$$\mathbb{P}(\varrho_r \geq \epsilon) \leq \frac{\mathbb{E}\varrho_r}{\epsilon} \leq \frac{(\frac{1}{2})^r[F(\widetilde{w}^r) - F(w^*)]}{\epsilon} \tag{46}$$

For a given $\zeta \in (0, 1)$, let the right side of (46) be not greater than $\zeta$, which requires

$$r \geq \log_2 \frac{F(\widetilde{w}^r) - F(w^*)}{\epsilon\zeta}. \tag{47}$$

Therefore, the result in (44) holds with probability at least $1 - \zeta$. Thus, we need $\mathcal{O}(\log(\frac{1}{\epsilon}))$ outer iterations to get $\widetilde{w}^r$ satisfying (44). Since within each outer-iteration, we need to calculate a full gradient and $m$ stochastic variance reduced gradients with mini-batch size $|\mathcal{B}|$, the overall oracle complexity is $\mathcal{O}((n + \frac{\kappa_{\widehat{s}}|\mathcal{B}|}{k})\log\frac{1}{\epsilon})$. Note that in high-dimensional region, the effect of support of snapshot point can be ignored. On the other hand, we only perform a hard thresholding operation at each outer iteration. Then we have

$$\log_2\left(\frac{1}{\epsilon}\right) = \frac{\log(\frac{1}{\epsilon})}{\log(2)} = \mathcal{O}(\log(\frac{1}{\epsilon})). \tag{48}$$

Thus, the hard thresholding oracle complexity is $\mathcal{O}(\log(\frac{1}{\epsilon}))$. This completes the proof. □

## D PROOF OF THEOREM OF SPARSE VARIANT

In this section, we theoretically analyze the convergence properties of our Serial Sparse SBCD-HTP (S²BCD-HTP).

---

**Algorithm 3** Serial Sparse SBCD-HTP (S$^2$BCD-HTP)

---

**Input:** number of outer-loops $R$, number of inner-loops $m$, step size $\eta$, sparsity level $s$;
**Initialize:** $\widetilde{w}^0$;
 1: **for** $r = 0, 1, \ldots, R - 1$ **do**
 2:   $w^0 = \widetilde{w} = \widetilde{w}^r$;
 3:   $\nabla F(\widetilde{w}) = \frac{1}{n} \sum_{i=1}^{n} \nabla f_i(\widetilde{w})$;
 4:   $\widetilde{\mathcal{G}} = \text{supp}(\widetilde{w})$;
 5:   **for** $t = 0, 1, \ldots, m - 1$ **do**
 6:     Randomly sample $i_t$ from $\{1, 2, \ldots, n\}$ uniformly;
 7:     Randomly sample $j_t$ from $\{1, 2, \ldots, k\}$ uniformly;
 8:     $T_{i_t} :=$ support of sample $i_t$;
 9:     $\mathcal{S} = \widetilde{\mathcal{G}} \cup \mathcal{G}_{j_t}$ ;
10:     $\nabla_{\mathcal{S}} g([w^t]_{T_{i_t}}) = \nabla_{\mathcal{S}} f_{i_t}([w^t]_{T_{i_t}}) - \nabla_{\mathcal{S}} f_{i_t}([\widetilde{w}]_{T_{i_t}}) + D_{i_t} \nabla_{\mathcal{S}} F(\widetilde{w})$;
11:     $w^{t+1} = w^t - \eta \nabla_{\mathcal{S}} g([w^t]_{T_{i_t}})$;
12:   **end for**
13:   $\widetilde{w}^{r+1} = HT(w^m, s)$;
14: **end for**
**Output:** $\widetilde{w}^R$.

---

### D.1  PROOF

*Proof.*    We start with the iterate difference between $w^{t+1}$ and $w^*$. By expanding iterate difference and taking expectation with respect to all randomness, we get

$$
\begin{aligned}
\mathbb{E}\|w^{t+1} - w^*\|^2 &= \mathbb{E}\|w^t - \eta \nabla_{\mathcal{S}} g_{\mathcal{I}}(w^t) - w^*\|^2 \\
&= \|w^t - w^*\|^2 + \eta^2 \|\nabla_{\mathcal{S}} g_{\mathcal{I}}(w^t)\|^2 + 2\eta \mathbb{E}\langle \nabla_{\mathcal{S}} g_{\mathcal{I}}(w^t), w^* - w^t \rangle \\
&\leq \|w^t - w^*\|^2 + \eta^2 \|\nabla_{\mathcal{S}} g_{\mathcal{I}}(w^t)\|^2 + \frac{2\eta}{k} \langle \nabla_{\mathcal{I}} F(w^t), w^* - w^t \rangle,
\end{aligned}
\tag{49}
$$

where the last inequality holds by the unbiasedness of the sparse gradient estimator $\nabla g_{\mathcal{I}}(w^t)$ and the expectation with respect to randomized block. By using the $\rho_{\widehat{s}}^-$-strongly convex condition, we get the bound for $\langle \nabla F(w^t), w^* - w^t \rangle$ as follows:

$$
\langle \nabla F(w^t), w^* - w^t \rangle \leq F(w^*) - F(w^t) - \frac{\rho_{\widehat{s}}^-}{2} \|w^* - w^t\|^2.
\tag{50}
$$

Due to the support set $\mathcal{I}$, we need the extended RSC condition with $\delta > 1$, which has been introduced in the proof of Lemma 5. Based on this underlying consensus, we can obtain the bound for $\langle \nabla F_{\mathcal{I}}(w^t), w^* - w^t \rangle$.

$$
\langle \nabla_{\mathcal{I}} F(w^t), w^* - w^t \rangle \leq \delta[F(w^*) - F(w^t)].
\tag{51}
$$

Using Lemma 6, we have the following sparse variance gradient bound,

$$
\begin{aligned}
\mathbb{E}[\|\nabla_{\mathcal{S}} g_{\mathcal{I}}(w^t)\|^2] \leq \frac{1}{k} \big( &12\delta \rho_{\widehat{s}}^+ [F(w^t) - F(w^*)] + (3 + 12(D_m^2 - 1))\|\nabla_{\mathcal{I}} F(w^*)\|^2 \\
&+ (24\delta \rho_{\widehat{s}}^+ + 48\delta \rho_{\widehat{s}}^+ (D_m^2 - 1))[F(\widetilde{w}) - F(w^*)] \big),
\end{aligned}
\tag{52}
$$

By combining the above inequalities, we have

$$
\begin{aligned}
\mathbb{E}\|w^{t+1} - w^*\|^2 &\leq \|w^t - w^*\|^2 + \eta^2 \|\nabla_{\mathcal{S}} g_{\mathcal{I}}(w^t)\|^2 - \frac{2\eta\delta}{k}[F(w^t) - F(w^*)] \\
&\leq \|w^t - w^*\|^2 + \frac{12\eta^2 \delta \rho_{\widehat{s}}^+ - 2\eta\delta}{k}[F(w^t) - F(w^*)] \\
&\quad + \frac{2\delta\eta^2}{k}(12\rho_{\widehat{s}}^+ + 24\rho_{\widehat{s}}^+ (D_m^2 - 1))[F(\widetilde{w}) - F(w^*)] \\
&\quad + \frac{\eta^2}{k}(3 + 12(D_m^2 - 1))\|\nabla_{\mathcal{I}} F(w^*)\|^2,
\end{aligned}
\tag{53}
$$

where the second inequality holds by Lemma 6. Summing the above inequality over $t = 0, \cdots, m-1$ and taking expectation with respect to all randomness in this epoch, we have

$$
\begin{aligned}
\mathbb{E}\|w^m - w^*\|^2 \leq{} & \mathbb{E}\|\widetilde{w}^{r-1} - w^*\|^2 - \frac{2\eta\delta m(1 - 6\eta\rho_{\widehat{s}}^+)}{k}[F(w^m) - F(w^*)] \\
& + \frac{2\delta\eta^2 m}{k}(12\rho_{\widehat{s}}^+ + 24\rho_{\widehat{s}}^+(D_m^2 - 1))[F(\widetilde{w}) - F(w^*)] \\
& + \frac{\eta^2 m}{k}(3 + 12(D_m^2 - 1))\|\nabla_{\mathcal{I}}F(w^*)\|^2.
\end{aligned}
\tag{54}
$$

Using (28), we obtain

$$
\begin{aligned}
\mathbb{E}\|w^m - w^*\|^2 \leq{} & \mathbb{E}\|\widetilde{w}^{r-1} - w^*\|^2 - \frac{2\eta\delta m}{k}(1 - 6\eta\rho_{\widehat{s}}^+)[F(\widetilde{w}^r) - F(w^*)] \\
& + 2\eta\delta m(1 - 6\eta\rho_{\widehat{s}}^+)\frac{\rho_{\widehat{s}}^+(d - s)}{2k\sigma(d - s^*)}\|w^m - w^*\|^2 \\
& + \frac{\delta\eta^2 m}{k}(24\rho_{\widehat{s}}^+ + 48\rho_{\widehat{s}}^+(D_m^2 - 1))[F(\widetilde{w}^{r-1}) - F(w^*)] \\
& + \frac{\eta^2 m}{k}(3 + 12(D_m^2 - 1))\|\nabla_{\mathcal{I}}F(w^*)\|^2.
\end{aligned}
\tag{55}
$$

Since $\widetilde{w}^r = HT(w^m, s)$, i.e., $\widetilde{w}^r$ is the best $s$-sparse approximation of $w^m$, and due to Lemma 2, then we have following result,

$$
\begin{aligned}
& \frac{2\eta\delta\alpha m}{k}(1 - 6\eta\rho_{\widehat{s}}^+)[F(\widetilde{w}^r) - F(w^*)] + (1 - 2\eta\delta m(1 - 6\eta\rho_{\widehat{s}}^+)\frac{\rho_{\widehat{s}}^+(d - s)}{2k\sigma(d - s^*)})\|\widetilde{w}^r - w^*\|^2 \\
& \leq \alpha\mathbb{E}\|\widetilde{w}^{r-1} - w^*\|^2 + \frac{\delta\eta^2 m\alpha}{k}(24\rho_{\widehat{s}}^+ + 48\rho_{\widehat{s}}^+(D_m^2 - 1))[F(\widetilde{w}^{r-1}) - F(w^*)] \\
& \quad + \frac{\eta^2 m\alpha}{k}(3 + 12(D_m^2 - 1))\|\nabla_{\mathcal{I}}F(w^*)\|^2 \\
& \leq \frac{2\alpha}{\rho_{\widehat{s}}^-}[F(\widetilde{w}^{r-1}) - F(w^*)] + \frac{\delta\eta^2 m\alpha}{k}(24\rho_{\widehat{s}}^+ + 48\rho_{\widehat{s}}^+(D_m^2 - 1))[F(\widetilde{w}^{r-1}) - F(w^*)] \\
& \quad + \frac{\eta^2 m\alpha}{k}(3 + 12(D_m^2 - 1))\|\nabla_{\mathcal{I}}F(w^*)\|^2,
\end{aligned}
\tag{56}
$$

where $\alpha = 1 + \frac{2\sqrt{s^*}}{\sqrt{s - s^*}}$, and we use $\rho_{\widehat{s}}^-$-strongly convex of $F(w)$ to bound $\|\widetilde{w}^{r-1} - w^*\|^2$ in the last inequality. Through proper simplification, we can obtain

$$
\begin{aligned}
& \frac{2\eta\alpha\delta m}{k}(1 - 6\eta\rho_{\widehat{s}}^+)[F(\widetilde{w}^r) - F(w^*)] \\
& \leq (\frac{2\alpha}{\rho_{\widehat{s}}^-} + \frac{(24\rho_{\widehat{s}}^+ + 48\rho_{\widehat{s}}^+(D_m^2 - 1))\delta\eta^2 m\alpha}{k})[F(\widetilde{w}^{r-1}) - F(w^*)] \\
& \quad + \frac{\eta^2 m\alpha}{k}(3 + 12(D_m^2 - 1))\|\nabla_{\mathcal{I}}F(w^*)\|^2,
\end{aligned}
\tag{57}
$$

where the second term of LHS of (56) is omitted due to the fact that the coefficient of $\mathbb{E}\|\widetilde{w}^r - w^*\|^2$ is larger than zero, when $6\rho_{\widehat{s}}^+ m\eta^2 - m\eta + \frac{k\sigma(d - s^*)}{\delta\rho_{\widehat{s}}^+(d - s)} > 0$, which can be naturally satisfied under our parameter setting with $s \geq \kappa_{\widehat{s}}s^*$. Therefore, we can obtain the following result,

$$
\begin{aligned}
\mathbb{E}[F(\widetilde{w}^r) - F(w^*)] \leq{} & \frac{\frac{2k\alpha}{\rho_{\widehat{s}}^-} + (24 + 48(D_m^2 - 1))\delta m\alpha\rho_{\widehat{s}}^+\eta^2}{2\eta\alpha\delta m(1 - 6\eta\rho_{\widehat{s}}^+)} \cdot [F(\widetilde{w}^{r-1}) - F(w^*)] \\
& + \frac{(3 + 12(D_m^2 - 1))m\alpha\eta^2}{2\eta\alpha\delta m(1 - 6\eta\rho_{\widehat{s}}^+)}\|\nabla_{\tilde{\mathcal{I}}}F(w^*)\|^2,
\end{aligned}
\tag{58}
$$

where $\tilde{\mathcal{I}} = \text{supp}(w^*) \cup \text{supp}(HT(\nabla F(w^*), 2s))$. We define $\xi := \frac{[3+12(D_m^2-1)]\eta}{2\delta(1-6\rho_{\widehat{s}}^+\eta)}\|\nabla_{\tilde{\mathcal{I}}}F(w^*)\|^2$.
In order to obtain a linear convergence rate, we have to bound

$$\beta := \frac{\frac{2k\alpha}{\rho_{\widehat{s}}^-} + (24 + 48(D_m^2-1))\delta m\alpha\rho_{\widehat{s}}^+\eta^2}{2\eta\alpha\delta m(1-6\eta\rho_{\widehat{s}}^+)} = \frac{k}{(\eta - 6\rho_{\widehat{s}}^+\eta^2)\delta m\rho_{\widehat{s}}^-} + \frac{(12 + 24(D_m^2-1))\rho_{\widehat{s}}^+\eta}{1-6\rho_{\widehat{s}}^+\eta} < 1. \tag{59}$$

If we choose $\eta \leq \frac{1}{48D_m^2\rho_{\widehat{s}}^+}$, then the second term of $\beta$ will be less than $\frac{1}{2}$, i.e., $\frac{(12+24(D_m^2-1))\rho_{\widehat{s}}^+\eta}{1-6\rho_{\widehat{s}}^+\eta} \leq \frac{1}{2}$. As for the first term, when $m \geq \frac{500D_m^4\kappa_{\widehat{s}}}{D_m^2-1}$ with $\eta \geq \frac{1}{50D_m^2\rho_{\widehat{s}}^+}, \delta = 4$ and $k = 10$, we have $\frac{1}{(\eta-6\rho_{\widehat{s}}^+\eta^2)\delta m\rho_{\widehat{s}}^-} \leq \frac{1}{4}$. Thus, we get the linear convergence for our serial sparse SBCD-HTP (S$^2$BCD-HTP).

$$\mathbb{E}[F(\widetilde{w}^r) - F(w^*)] \leq \frac{3}{4} \cdot [F(\widetilde{w}^{r-1}) - F(w^*)] + \xi, \tag{60}$$

which means that the total gradient evaluation oracle complexity is $\mathcal{O}((n + \frac{\kappa_{\widehat{s}}}{k}\log\frac{1}{\epsilon})$, and the hard thresholding oracle complexity is $\mathcal{O}(\log(\frac{1}{\epsilon}))$. This completes the proof. $\square$

# E    PROOF OF THEOREM 2

In this section, we analyze the convergence properties of the proposed Asynchronous Sparse SBCD-HTP (ASBCD-HTP). We first have to specify the iterates labeling order used in our asynchronous analysis.

We use "After Read" labeling order (Leblond et al., 2017) in our proof, which enjoys a simpler analysis but requires the order of randomly sampling step to be an uniform distributed sample. In order to give a clear proof, we adopt the "After Read" labeling order and make the following assumptions:
"After Read" labeling order:
1. Inconsistently read the iterate $\hat{w}_t$;
2. Increase iterates counter $t + 1$ and sample a random sample $i_{t+1}$;
3. Compute the update $-\eta\nabla g(\hat{w}_t)$;
4. Atomic write the update to shared variable coordinately.

**Assumption 4** *The labeling order increases after Step 8 in Algorithm 2 finished, and thus the future perturbation is not considered in the effect of asynchrony in the current step.*

**Assumption 5** *We assume that uniform distributed samples and the independence of the sample $i_{t+1}$ with $\hat{w}_t$.*

Following (Leblond et al., 2017), we can explicit the effect of asynchrony as follows:

$$\hat{w}_t - w^t = \eta \sum_{u=(t-\tau)_+}^{t-1} G_u\nabla g(\hat{w}_u), \tag{61}$$

where $w^t$ is the $t$-th inner iteration solution. In the proof of Theorem 2, we use $w^t$ to represent the temporary solution of the $t$-th iteration for a clean representation. And $G_u$ is a diagonal matrix with entries in $\{0, +1\}$. This explicitly defines the coordinate perturbation from the past updates. Here, $\tau$ denotes the maximum number of overlaps between concurrent threads. We also denote $\Delta = \max_{j=1,\cdots,d} p_j$, which provides a measure of sparsity following (Leblond et al., 2017).

## E.1    PROOF

*Proof.* We analyze our asynchronous algorithm based on the "perturbed iterate analysis" framework (Mania et al., 2017) with "After Read" labeling order. By expanding the iterate difference and taking expectation with respect to the sample $i_{t+1}$, we get

$$
\begin{aligned}
\mathbb{E}_{i_{t+1}}\|w^{t+1} - w^*\|^2 &= \mathbb{E}_{i_{t+1}}\|w^t - \eta\nabla_{\mathcal{S}}g_{\mathcal{I}}(\hat{w}_t) - w^*\|^2 \\
&= \|w^t - w^*\|^2 + \eta^2\|\nabla_{\mathcal{S}}g_{\mathcal{I}}(\hat{w}_t)\|^2 + 2\eta\mathbb{E}_{i_{t+1}}\langle w^* - w^t, \nabla_{\mathcal{S}}g_{\mathcal{I}}(\hat{w}_t)\rangle \\
&\leq \|w^t - w^*\|^2 + \eta^2\|\nabla_{\mathcal{S}}g_{\mathcal{I}}(\hat{w}_t)\|^2 + \frac{2\eta}{k}\mathbb{E}_{i_{t+1}}\langle w^* - \hat{w}_t, \nabla g_{\mathcal{I}}(\hat{w}_t)\rangle \\
&\quad + 2\eta\mathbb{E}_{i_{t+1}}\langle \hat{w}_t - w^t, \nabla_{\mathcal{S}}g_{\mathcal{I}}(\hat{w}_t)\rangle,
\end{aligned}
\tag{62}
$$

where $\hat{w}_t$ is the "perturbed" iterate with perturbation $\psi$ and the last inequality follows from the expectation with respect to randomized block. We need to bound the term, $2\eta\langle w^* - \hat{w}_t, \nabla g_{\mathcal{I}}(\hat{w}_t)\rangle$. Since $i_{t+1}$ is independent to $\hat{w}_t$, we can write:

$$
\mathbb{E}_{i_{t+1}}\langle w^* - \hat{w}_t, \nabla g_{\mathcal{I}}(\hat{w}_t)\rangle = \langle w^* - \hat{w}_t, \mathbb{E}_{i_{t+1}}\nabla g_{\mathcal{I}}(\hat{w}_t)\rangle = \langle w^* - \hat{w}_t, \nabla_{\mathcal{I}}F(\hat{w}_t)\rangle.
\tag{63}
$$

We can use $\rho_{\hat{s}}^-$-strong convexity bound of $F(\hat{w})$ with constant factor $\delta > 1$ as well as a squared triangle inequality to get:

$$
-\langle \hat{w}_t - w^*, \nabla F(\hat{w}_t)\rangle \leq -\delta(F(\hat{w}_t) - F(w^*)) - \frac{\rho_{\hat{s}}^+}{2}\|\hat{w}_t - w^*\|^2,
\tag{64}
$$

$$
-\|\hat{w}_t - w^*\|^2 \leq \|\hat{w}_t - w^t\|^2 - \frac{1}{2}\|w^t - w^*\|^2, \quad (\|a + b\|^2 \leq 2\|a\|^2 + 2\|b\|^2),
\tag{65}
$$

$$
\langle w^* - \hat{w}_t, \nabla_{\mathcal{I}}F(\hat{w}_t)\rangle \leq -\delta(F(\hat{w}_t) - F(w^*)) + \frac{\rho_{\hat{s}}^-}{2}\|\hat{w}_t - w^t\|^2 - \frac{\rho_{\hat{s}}^-}{4}\|w^t - w^*\|^2.
\tag{66}
$$

Then we make an identical deformation, and we have

$$
\begin{aligned}
\frac{2\eta}{k}\mathbb{E}_{i_{t+1}}\langle w^* - \hat{w}_t, \nabla g_{\mathcal{I}}(\hat{w}_t)\rangle &\leq -\frac{\eta\rho_{\hat{s}}^-}{2k}\mathbb{E}_{i_{t+1}}\|w^t - w^*\|^2 + \frac{\eta\rho_{\hat{s}}^-}{k}\mathbb{E}_{i_{t+1}}\|\hat{w}_t - w^t\|^2 \\
&\quad - \frac{2\eta\delta}{k}(F(\hat{w}_t) - F(w^*)).
\end{aligned}
\tag{67}
$$

Putting it all together, we get the initial recursive inequity, written here explicitly:

$$
\begin{aligned}
a^{t+1} &\leq (1 - \frac{\eta\rho_{\hat{s}}^-}{2k})a^t + \eta^2\mathbb{E}_{i_{t+1}}\|\nabla_{\mathcal{S}}g_{\mathcal{I}}(\hat{w}_t)\|^2 + \frac{\eta\rho_{\hat{s}}^-}{k}\mathbb{E}_{i_{t+1}}\|\hat{w}_t - w^t\|^2 \\
&\quad + 2\eta\mathbb{E}_{i_{t+1}}\langle \hat{w}_t - w^t, \nabla_{\mathcal{S}}g_{\mathcal{I}}(\hat{w}_t)\rangle - \frac{2\eta\delta}{k}\hat{e}^t,
\end{aligned}
\tag{68}
$$

where $a^t := \mathbb{E}_{i_{t+1}}\|w^t - w^*\|^2$ and $\hat{e}^t := \mathbb{E}_{i_{t+1}}[F(\hat{w}_t) - F(w^*)]$.

From Lemma 1 in (Leblond et al., 2017), we can bound the asynchronous terms $\|\hat{w}_t - w^t\|^2$, $\langle \hat{w}_t - w^t$ and $\nabla g_{\mathcal{I}}(\hat{w}_t)\rangle$ as follows.

$$
\begin{aligned}
\mathbb{E}\langle \hat{w}_t - w^t, \nabla_{\mathcal{S}}g_{\mathcal{I}}(\hat{w}_t)\rangle &= \eta\sum_{u=(t-\tau)_+}^{t-1}\mathbb{E}\langle G_u\nabla_{\mathcal{S}}g_{\mathcal{I}}(\hat{w}_u), \nabla_{\mathcal{S}}g_{\mathcal{I}}(\hat{w}_u)\rangle \\
&\leq \eta\sum_{u=(t-\tau)_+}^{t-1}\mathbb{E}|\langle \nabla_{\mathcal{S}}g_{\mathcal{I}}(\hat{w}_u), \nabla_{\mathcal{S}}g_{\mathcal{I}}(\hat{w}_u)\rangle| \\
&\leq \eta\sum_{u=(t-\tau)_+}^{t-1}\frac{\sqrt{\Delta}}{2}(\mathbb{E}\|\nabla_{\mathcal{S}}g_{\mathcal{I}}(\hat{w}_u)\|^2 + \mathbb{E}\|\nabla_{\mathcal{S}}g_{\mathcal{I}}(\hat{w}_t)\|^2) \\
&\leq \frac{\eta\sqrt{\Delta}}{2}\sum_{u=(t-\tau)_+}^{t-1}\mathbb{E}\|\nabla_{\mathcal{S}}g_{\mathcal{I}}(\hat{w}_u)\|^2 + \frac{\eta\sqrt{\Delta}}{2}\mathbb{E}\|\nabla_{\mathcal{S}}g_{\mathcal{I}}(\hat{w}_t)\|^2.
\end{aligned}
\tag{69}
$$

$$\mathbb{E}\|\hat{w}_t - w^t\|^2 \leq \sum_{v,u=(t-\tau)_+}^{t-1} |\langle G_u \nabla_{\mathcal{S}} g_{\mathcal{I}}(\hat{w}_u), G_v \nabla_{\mathcal{S}} g_{\mathcal{I}}(\hat{w}_v) \rangle|$$

$$\leq \eta^2 \sum_{u=(t-\tau)_+}^{t-1} \|\nabla_{\mathcal{S}} g_{\mathcal{I}}(\hat{w}_u)\|^2 + \eta^2 \sum_{u,v=(t-\tau)_+, u\neq v}^{t-1} |\langle G_u \nabla_{\mathcal{S}} g_{\mathcal{I}}(\hat{w}_u), G_v \nabla_{\mathcal{S}} g_{\mathcal{I}}(\hat{w}_v) \rangle|$$

$$\leq \eta^2 \sum_{u=(t-\tau)_+}^{t-1} \mathbb{E}\|\nabla_{\mathcal{S}} g_{\mathcal{I}}(\hat{w}_u)\|^2 + \eta^2 \sqrt{\Delta}(\tau-1)_+ \sum_{u=(t-\tau)_+}^{t-1} \mathbb{E}\|\nabla_{\mathcal{S}} g_{\mathcal{I}}(\hat{w}_u)\|^2$$

$$= \eta^2 (1 + \sqrt{\Delta}(\tau-1)_+) \sum_{u=(t-\tau)_+}^{t-1} \mathbb{E}\|\nabla_{\mathcal{S}} g_{\mathcal{I}}(\hat{w}_u)\|^2$$

$$\leq \eta^2 (1 + \sqrt{\Delta}\tau) \sum_{u=(t-\tau)_+}^{t-1} \mathbb{E}\|\nabla_{\mathcal{S}} g_{\mathcal{I}}(\hat{w}_u)\|^2, \tag{70}$$

where $G_u$ are $d \times d$ diagonal matrices whose entries in $\{0, +1\}$. Each update in $\hat{w}_t$ is already in $w^t$– this is the case of $0$. On the contrary, some updates might be late: this is the case of $+1$. Then by combining (68), (69) and (70), we get

$$a^{t+1} \leq (1 - \frac{\eta \rho_{\hat{s}}^-}{2k})a^t + \eta^2\|\nabla_{\mathcal{S}} g_{\mathcal{I}}(\hat{w}_t)\|^2 + \eta^3 \rho_{\hat{s}}^-(1 + \sqrt{\Delta}\tau) \sum_{u=(t-\tau)_+}^{t-1} \mathbb{E}_{i_{t+1}}\|\nabla_{\mathcal{S}} g_{\mathcal{I}}(\hat{w}_u)\|^2$$

$$+ \eta^2 \sqrt{\Delta} \sum_{u=(t-\tau)_+}^{t-1} \mathbb{E}_{i_{t+1}}\|\nabla_{\mathcal{S}} g_{\mathcal{I}}(\hat{w}_u)\|^2 + \eta^2 \sqrt{\Delta}\tau \mathbb{E}_{i_{t+1}}\|\nabla_{\mathcal{S}} g_{\mathcal{I}}(\hat{w}_t)\|^2 - \frac{2\eta\delta}{k}\hat{e}^t$$

$$\leq (1 - \frac{\eta \rho_{\hat{s}}^-}{2k})a^t + \eta^2 C_1 \mathbb{E}_{i_{t+1}}\|\nabla_{\mathcal{S}} g_{\mathcal{I}}(\hat{w}_t)\|^2 + \eta^2 C_2 \sum_{u=(t-\tau)_+}^{t-1} \mathbb{E}_{i_{t+1}}\|\nabla_{\mathcal{S}} g_{\mathcal{I}}(\hat{w}_u)\|^2 - \frac{2\eta\delta}{k}\hat{e}^t, \tag{71}$$

where $C_1 := 1 + \sqrt{\Delta}\tau$ and $C_2 := \sqrt{\Delta} + \eta \rho_{\hat{s}}^- C_1$. $\tau$ represents the mawimum number of overlaps between concurrent threads. And $\Delta = \max_{j=1,\cdots,d} p_j$, which is a measure of sparsity with $1/n \leq \Delta \leq 1$.

Using Lemma 6 to bound the sparse variance term, the above inequality becomes

$$a^{t+1} \leq (1 - \frac{\eta \rho_{\hat{s}}^-}{2k})a^t + \frac{\eta^2 C_1}{k}(12\delta \rho_{\hat{s}}^+[F(\hat{w}_t) - F(w^*)] + (24 + 48(D_m^2 - 1))\delta \rho_{\hat{s}}^+[F(\widetilde{w}) - F(w^*)]$$

$$+ (3 + 12(D_m^2 - 1))\|\nabla_{\mathcal{I}} F(w^*)\|^2) + \frac{\eta^2 C_2}{k} \sum_{u=(t-\tau)_+}^{t-1} \mathbb{E}_{i_{t+1}}\|\nabla g_{\mathcal{I}}(\hat{w}_u)\|^2 - \frac{2\eta\delta}{k}\hat{e}^t$$

$$\leq a^t + \frac{\eta^2 C_2}{k} \sum_{u=(t-\tau)_+}^{t-1} \mathbb{E}_{i_{t+1}}\|\nabla g_{\mathcal{I}}(\hat{w}_u)\|^2 - \frac{2\eta\delta(1 - 6\rho_{\hat{s}}^+ \eta C_1)}{k}\hat{e}^t$$

$$+ (24\rho_{\hat{s}}^+ + 48\rho_{\hat{s}}^+(D_m^2 - 1))\frac{\delta C_1 \eta^2}{k}\tilde{e} + (3 + 12(D_m^2 - 1))\frac{\eta^2 C_1}{k}\|\nabla_{\mathcal{I}} F(w^*)\|^2, \tag{72}$$

where $\tilde{e} := \mathbb{E}[F(\widetilde{w}) - F(w^*)]$, $\hat{e}^t := \mathbb{E}[F(\hat{w}_t) - F(w^*)]$. Note that we have $\widetilde{w} = w^0 = \widetilde{w}^{r-1}$. Summing (72) over $t = 0, \cdots, m-1$ and taking expectation with all randomness in this epoch, we get

$$a^m \leq a^0 - \frac{2\eta\delta m}{k}(1 - 6\rho_{\hat{s}}^+ \eta C_1)\hat{e}^m + \frac{\eta^2 C_2}{k} \sum_{t=1}^{m} \sum_{u=(t-\tau)_+}^{t-1} \mathbb{E}[\|\nabla g_{\mathcal{I}}(\hat{w}_u)\|^2]$$

$$+ (24\rho_{\hat{s}}^+ + 48\rho_{\hat{s}}^+(D_m^2 - 1))\frac{\delta m C_1 \eta^2}{k}\tilde{e}^{r-1} + (3 + 12(D_m^2 - 1))\frac{m\eta^2 C_1}{k}\|\nabla_{\mathcal{I}} F(w^*)\|^2, \tag{73}$$

where $a^m = \mathbb{E}\|w^m - w^*\|^2$, and $a^0 = \mathbb{E}\|w^0 - w^*\|^2$. Then we focus on upper bounding the third term on the RHS of (73),

$$
\sum_{t=1}^{m} \sum_{u=(t-\tau)_+}^{t-1} \mathbb{E}[\|\nabla g_{\mathcal{I}}(\hat{w}_u)\|^2] \le \tau \sum_{t=1}^{m-1} \mathbb{E}[\|\nabla g_{\mathcal{I}}(\hat{w}_t)\|^2] \le \tau \sum_{t=1}^{m} \mathbb{E}[\|\nabla g_{\mathcal{I}}(\hat{w}_t)\|^2]
$$
$$
\le \tau(12\rho_{\hat{s}}^+ \delta m \hat{e}^m + \delta m(24\rho_{\hat{s}}^+ + 48\rho_{\hat{s}}^+(D_m^2 - 1))\tilde{e}^{r-1} + m(3 + 12(D_m^2 - 1))\|\nabla_{\mathcal{I}} F(w^*)\|^2). \tag{74}
$$

Substituting the above inequality into (73), we obtain

$$
a^m \le a^0 - \frac{2\eta\delta m}{k}(1 - 6\rho_{\hat{s}}^+ \eta(\tau C_2 + C_1))\hat{e}^m + \frac{\delta\rho_{\hat{s}}^+ m\eta^2}{k}(24 + 48(D_m^2 - 1))(\tau C_2 + C_1)\tilde{e}^{r-1}
$$
$$
+ \frac{\eta^2 m}{k}(3 + 12(D_m^2 - 1))(C_1 + \tau C_2)\|\nabla_{\mathcal{I}} F(w^*)\|^2. \tag{75}
$$

Then we bring (28) into the above inequality, we have

$$
a^m \le a^0 - \frac{2\eta\delta m}{k}(1 - 6\rho_{\hat{s}}^+ \eta(\tau C_2 + C_1))\mathbb{E}[F(\hat{w}_m) - F(\widetilde{w}^r) + F(\widetilde{w}^r) - F(w^*)]
$$
$$
+ \frac{\delta\rho_{\hat{s}}^+ m\eta^2}{k}(24 + 48(D_m^2 - 1))(\tau C_2 + C_1)\tilde{e}^{r-1}
$$
$$
+ \frac{\eta^2 m}{k}(3 + 12(D_m^2 - 1))(C_1 + \tau C_2)\|\nabla_{\mathcal{I}} F(w^*)\|^2
$$
$$
\le a^0 - \frac{2\eta\delta m}{k}(1 - 6\rho_{\hat{s}}^+ \eta(\tau C_2 + C_1))\tilde{e}^r + 2\eta\delta m(1 - 6\rho_{\hat{s}}^+ \eta(\tau C_2 + C_1))\frac{\rho_{\hat{s}}^+(d - s)}{2k\sigma(d - s^*)}a^m
$$
$$
+ \frac{\delta\rho_{\hat{s}}^+ m\eta^2}{k}(24 + 48(D_m^2 - 1))(\tau C_2 + C_1)\tilde{e}^{r-1}
$$
$$
+ \frac{\eta^2 m}{k}(3 + 12(D_m^2 - 1))(C_1 + \tau C_2)\|\nabla_{\mathcal{I}} F(w^*)\|^2, \tag{76}
$$

where the second inequality holds since we use (28) and make a proper simplification. Using Lemma 2, we have

$$
(1 - 2\eta\delta m(1 - 6\rho_{\hat{s}}^+ \eta(\tau C_2 + C_1))\frac{\rho_{\hat{s}}^+(d - s)}{2k\sigma(d - s^*)})\tilde{a}^r
$$
$$
\le \frac{2\alpha}{\rho_{\hat{s}}^-}\tilde{e}^{r-1} - \frac{2\alpha\eta\delta m}{k}(1 - 6\rho_{\hat{s}}^+ \eta(\tau C_2 + C_1))\tilde{e}^r
$$
$$
+ \frac{\alpha\delta\rho_{\hat{s}}^+ m\eta^2}{k}(24 + 48(D_m^2 - 1))(\tau C_2 + C_1)\tilde{e}^{r-1}
$$
$$
+ \frac{\alpha\eta^2 m}{k}(3 + 12(D_m^2 - 1))(C_1 + \tau C_2)\|\nabla_{\tilde{\mathcal{I}}} F(w^*)\|^2, \tag{77}
$$

where $\alpha = 1 + \frac{2\sqrt{s^*}}{\sqrt{s - s^*}}$, $\tilde{a}^r = \|\widetilde{w}^r - w^*\|^2$, and $\tilde{\mathcal{I}} = \text{supp}(w^*) \cup \text{supp}(HT(\nabla F(w^*), 2s))$. The first term of LHS of (77) is omitted due to the coefficient of $\tilde{a}^r$ is larger than zero, when $6\rho_{\hat{s}}^+ m(\tau C_2 + C_1)\eta^2 - m\eta + \frac{k\sigma(d - s^*)}{\delta\rho_{\hat{s}}^+(d - s)} > 0$, which can be naturally satisfied under our parameter settings with $s \ge \kappa_{\hat{s}} s^*$. Therefore, we obtain the following recursive inequality,

$$
\tilde{e}^r \le \frac{\frac{2k\alpha}{\rho_{\hat{s}}^-} + \alpha\delta\rho_{\hat{s}}^+ m\eta^2(24 + 48(D_m^2 - 1))(\tau C_2 + C_1)}{2\alpha\eta\delta m(1 - 6\rho_{\hat{s}}^+ \eta(\tau C_2 + C_1))} \cdot \tilde{e}^{r-1}
$$
$$
+ \frac{\alpha\eta^2 m(3 + 12(D_m^2 - 1))(\tau C_2 + C_1)}{2\alpha\eta\delta m(1 - 6\rho_{\hat{s}}^+ \eta(\tau C_2 + C_1))} \cdot \|\nabla_{\tilde{\mathcal{I}}} F(w^*)\|^2. \tag{78}
$$

We assume that the following settings has satisfied with the above constraint and then by choosing $m = 120\kappa_{\widehat{s}}, \eta = \frac{1}{60\rho_{\widehat{s}}^+}, k = 10$, we get

$$\tilde{e}^r \leq \frac{20 + \frac{\delta}{75(\rho_{\widehat{s}}^+)^2}(1 + 2(D_m^2 - 1))(\tau C_2 + C_1)}{4\delta(1 - \frac{1}{10}(\tau C_2 + C_1))} \cdot \tilde{e}^{r-1} + \xi \cdot \|\nabla_{\widetilde{\mathcal{I}}} F(w^*)\|^2, \qquad (79)$$

where $\xi = \frac{\eta(3 + 12(D_m^2 - 1))(\tau C_2 + C_1)}{2\delta(1 - 6\rho_{\widehat{s}}^+ \eta(\tau C_2 + C_1))}$. In order to ensure linear speed up, $\tau$ needs to satisfy the following constraint,

$$\theta \triangleq \frac{20 + \frac{\delta}{75(\rho_{\widehat{s}}^+)^2}(1 + 2(D_m^2 - 1))(\tau C_2 + C_1)}{4\delta(1 - \frac{1}{10}(\tau C_2 + C_1))} \leq 1. \qquad (80)$$

We assume that $37(\rho_{\widehat{s}}^+)^2 \geq D_m^2$ and $\delta = 10$, then by simply setting $\tau \leq \min\{\frac{3}{5\sqrt{\Delta}}, 2\kappa_{\widehat{s}}, \sqrt{\frac{2\kappa_{\widehat{s}}}{\sqrt{\Delta}}}\}$, the above constraint is satisfied with $\theta \leq 0.7538$, which implies that the oracle gradient evaluation and hard thresholding oracle complexities are $\mathcal{O}((n + \frac{\kappa_{\widehat{s}}}{k})\log\frac{1}{\epsilon})$ and $\mathcal{O}(\log(\frac{1}{\epsilon}))$, respectively. We can simply infer the statistical estimation result as before. Here, we will omit this part. This completes the proof. □

## E.2 DISCUSSION ABOUT SPARSE VARIANCE BOUND

In the dense update case (i.e., $D_m = 1$), Lemma 6 is very similar to Lemma 5 when $|\mathcal{B}| = 1$. In the sparse update case, Lemma 6 highly correlates with the sparsity of datasets ($\propto D_m^2$), which could be much loose in some extreme cases. If $D_m$ is very large (imaging a dataset with some dimensions contain only one entry among the $n$ samples, so $D_m = n$), the step size of sparse (asynchronous) SBCD-HTP will be much smaller than that of SBCD-HTP. Our algorithms can still converge to the optimal solution up to the statistical error. Actually, it is still an open problem whether we can have a tighter variance bound in the sparse update setting that is uncorrelated with $D_m$.

## F   MORE EXPERIMENTAL RESULTS

Table 2: Summary of large-scale and high-dimensional datasets

| Datasets | # Data | # Feature | Density |
| --- | --- | --- | --- |
| rcv1-train | 20,242 | 47,236 | 0.16% |
| rcv1-test | 677,399 | 47,236 | 0.15% |
| real-sim | 72,309 | 20,958 | 0.024% |
| news20 | 19,996 | 30,000 | 0.49% |
| E2006-TFIDF | 16,087 | 80,000 | 1.54% |

## F.1   REAL-WORLD DATASETS AND EQUIPMENTS

We summarize the detailed information of all the real-world datasets in Table 2, including rcv1-train, rcv1-test, real-sim, news20 and E2006-TFIDF. All these datasets are provided in the LibSVM website[1]. The E2006-TFIDF dataset includes 16,087 training data points and 150,360 features.

---

[1] https://www.csie.ntu.edu.tw/~cjlin/libsvm/

Instead of using all features, we randomly select 80,000 features for training. As for the news20 dataset, we randomly choose 30,000 features for training.

### F.2 BASELINE METHODS

We compare our serial dense SBCD-HTP method with the state-of-the-art methods:

- **Fast Gradient with Hard Thresholding** (FG-HT) (Yuan et al., 2014): This method is based on a standard gradient descent step and combined with hard thresholding to solve sparsity-constrained problems.

- **Stochastic Gradient with Hard Thresholding** (SG-HT) (Nguyen et al., 2017): This method combines stochastic gradient descent with hard thresholding for solving large-scale sparsity-constrained problems.

- **Stochastic Variance Reduced Gradient with Hard Thresholding** (SVRG-HT) (Li et al., 2016b): This method incorporates the variance reduction technique in (Johnson & Zhang, 2013) into the hard thresholding algorithm to improve the efficiency of stochastic optimization.

- **Accelerated Stochastic Block Coordinate Gradient Descent with Hard Thresholding** (ASBCDHT) (Chen & Gu, 2016): Block coordinate descent (BCD) is used for solving sparsity-constrained problems.

- **Fast Newton Hard Thresholding Pursuit** (FNHTP) (Chen & Gu, 2017): This method tries to iteratively approximate the inverse Hessian matrix and combines a Newton algorithm with hard thresholding.

For the proposed Serial Sparse SBCD-HTP ($S^2$BCD-HTP) and Asynchronous sparse parallel SBCD-HTP (ASBCD-HTP) methods, we compare them with the following asynchronous parallel sparsity-constrained methods:

- **Asynchronous Accelerated Stochastic Block Coordinate Descent Hard Thresholding** ($A^2$SBCD-HT): This algorithm is the asynchronous variant of ASBCDHT by using our parallel framework.

- **Asynchronous Stochastic Gradient with Hard Thresholding** (ASG-HT): This method is an extension of SG-HT from serial dense setting to asynchronous sparse setting.

- **Asynchronous Stochastic Variance Reduced Gradient Hard Thresholding** (ASVRG-HT) (Li et al., 2016a): This method incorporates multicore structure to compute stochastic variance reduced gradient in a parallel way.

### F.3 FACE RECOGNITION DATASETS

Although there are many datasets available for face recognition, we choose a common dataset (i.e. the Extended Yale B database (Georghiades et al., 2001)). The Extended Yale B database contains 2,414 frontal-face images of 38 people under different controlled lighting conditions (Georghiades et al., 2001). For each individual, we randomly choose 26 images for training and 15 images for testing.

### F.4 EXPERIMENTAL SETUP

For each sparsity-constrained algorithm, the sparse level makes great difference to the solution of sparse representation, especially at different noise levels. Therefore, in order to approach the best performance of all these algorithms, we change the sparsity parameter within a certain range for each algorithm. Thus, we can make sure that all these algorithms achieve the best recognition rates in the parameter setting. In all settings of the experiments, the images are down-sampled to $32 \times 32$ pixels.

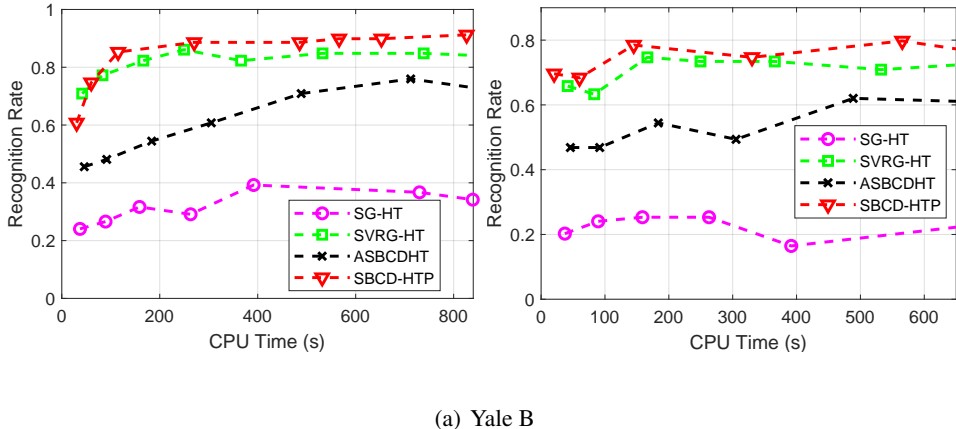

(a) Yale B

Figure 5: Recognition rates of all the algorithms on the Extended Yale B with different levels of Gaussian noise:

$\vartheta = 0.3$ (left) and $\vartheta = 0.6$ (right).

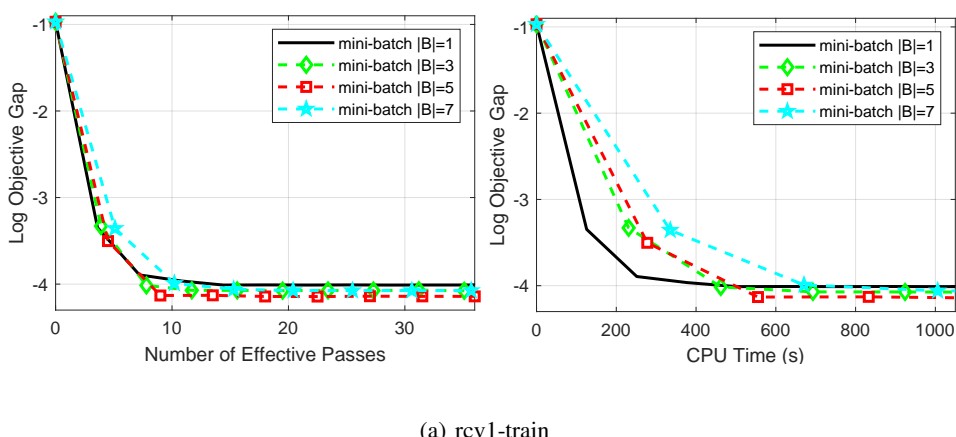

(a) rcv1-train

Figure 6: Mini-batch size in the convergence rate of SBCD-HTP algorithm for sparse logistic regression on

rcv1-train data.

Based on the sparse representation-based classification algorithm (SRC) (Wright et al., 2008), a series of processing operations are made to the above dataset. We first rescale the training matrix into [0,1] for the convenience of adding noise, and then add Gaussian noise with zero mean and standard deviation $\vartheta$. Finally, we normalize the columns of the training matrix to have unit $\ell_2$-norm. The parameter settings of comparison algorithms for face recognition task are the same as those described in Section 6.1.

## F.5 RESULTS

We first measure the effect of mini-batch size $|\mathcal{B}|$ in the convergence of SBCD-HTP on rcv1-train, as shown in Figure 6. It is obvious that the large mini-batch size can accelerate the convergence rate and make SBCD-HTP to obtain a better solution. This result is exactly in line with our theoretical analysis. However, when the mini-batch size is relatively large, it will deteriorate the performance of running time. One can simply choose the mini-batch size to 3 or 5 as suggested in Figure 6.

## F.6 FACE RECOGNITION TASKS

Then, we compare the performance of some hard thresholding algorithms, including SG-HT, SVRG-HT and ASBCDHT, with our SBCD-HTP for solving face recognition tasks on the Extended Yale B dataset. As shown in Figure 5, SBCD-HTP can achieve the best performance among these stochastic hard thresholding methods in a short time. It is clear that SBCD-HTP can not only optimize the objective function efficiently but also obtain a better representative solution on real applications, which is consistent with our theoretical results as described in Section 4.2. This result also shows that our SBCD-HTP has a better generalization accuracy than other hard thresholding algorithms.

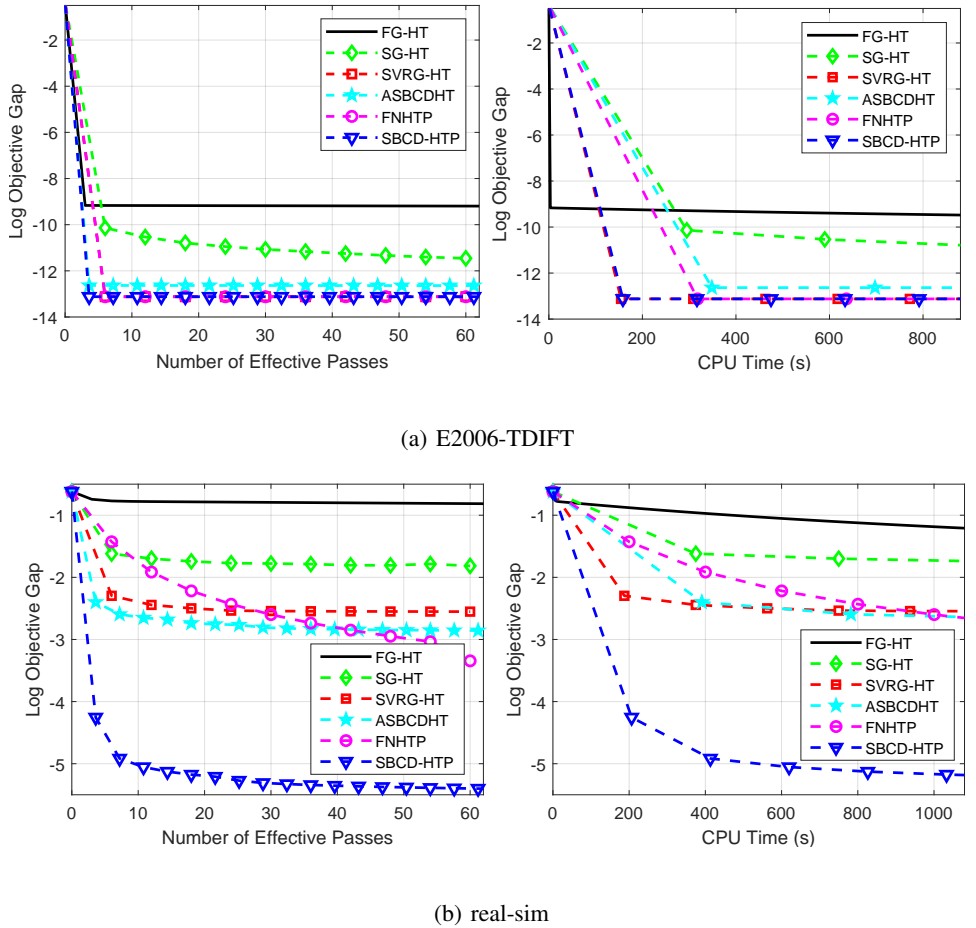

(a) E2006-TDIFT

(b) real-sim

Figure 7: Comparison of all the algorithms for solving sparse logistic regression problems. In each plot, the vertical axis shows the objective value minus the minimum, and the horizontal axis is the number of effective passes over data or running time (seconds).

## F.7 LINEAR/LOGISTIC REGISTRATION

We show more practical evaluation results for sparse linear regression and sparse logistic regression problems. All hardware environment and parameter settings are the same as in Section 6. For SBCD-HTP, from Figures 7 and 8, we can come to the same conclusion as we did before, that is in both sparse linear regression and sparse logistic regression, our algorithm converges faster than the baseline algorithms in terms of effective passes and running time. However, we find an interesting

result, which is that for the E2006-TFIDF dataset, all the algorithms converge with high precision solutions, so that we have to use some methods to magnify their differences. Of course, our SBCD-HTP has the best performance, but the gap is not that big. We think that this is due to ease with which E2006-TFIDF dataset can be optimized, since this phenomenon does not occur on other datasets.

For S$^2$BCD-HTP and ASBCD-HTP, from Figure 9, our S$^2$BCD-HTP and ASBCD-HTP far exceed the baseline algorithms in terms of running time. Therefore, we can confirmedly obtain the conclusion that our Sparse and Asynchronous algorithms outperform the state-of-the-art hard thresholding algorithms for sparse datasets. This is a significant improvement for hard thresholding algorithms to solve sparse learning problems in the high-dimensional asynchronous setting.

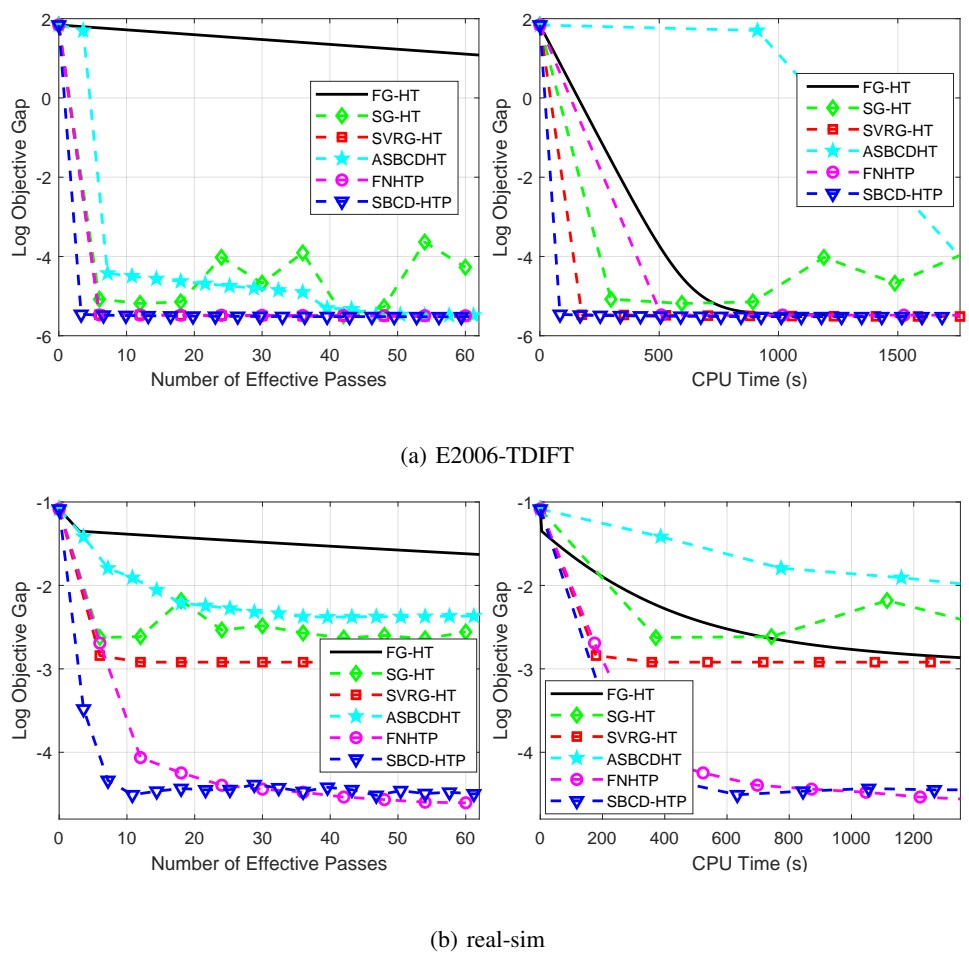

(a) E2006-TDIFT

(b) real-sim

Figure 8: Comparison of all the algorithms for solving sparse linear regression problems.

### F.7.1 UPPER BOUND

In order to conduct an experiment on the upper bound of Theorem 1. We first generate some $n \times d$ synthetic matrices $X$, each row of which is drawn independently from a $d$-dimensional Gaussian distribution with mean 0 and covariance matrix $\Sigma \in \mathbb{R}^{d \times d}$. The response vector is generated from the model $y = Xw^* + \epsilon$, where $w^* \in \mathbb{R}^d$ is the $s^*$-sparse coefficient vector, and we need to generate the noise $\epsilon$ drawn from a multivariate normal distribution $N(0, \sigma^2 I)$ with $\sigma^2 = 0.01$. The nonzero entries in $w^*$ are sampled independently from a uniform distribution over the interval $[-1, 1]$. For the experiments, we construct the following synthetic data set: $n = 5000$, $d = 10000$, $s^* = 500$,

and the diagonal entries of the covariance matrix $\Sigma$ are set to 1, and the other entries are set to 0.1.

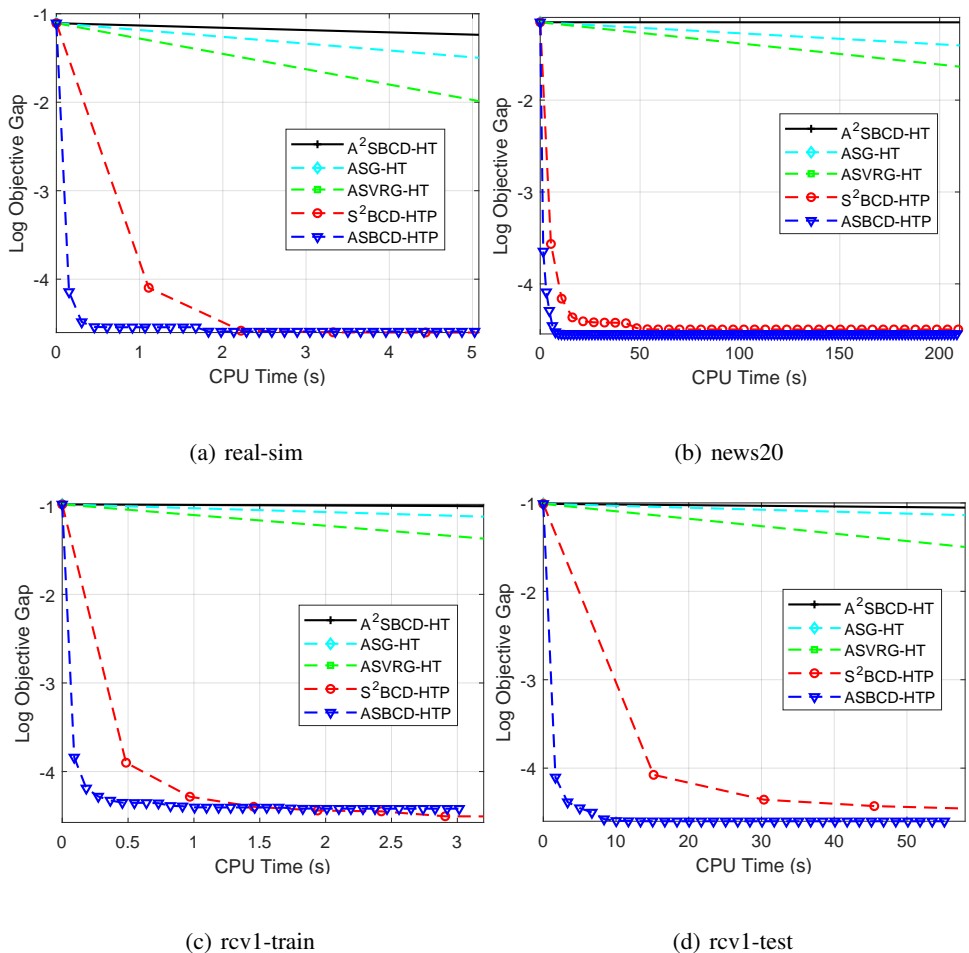

(a) real-sim

(b) news20

(c) rcv1-train

(d) rcv1-test

Figure 9: Comparison of ASG-HT, ASVRH-HT, A$^2$SBCD-HT, S$^2$BCD-HTP and ASBCD-HTP for solving

sparse linear regression problems. The four asynchronous parallel algorithms (i.e., ASG-HT, ASVRH-HT,

A$^2$SBCD-HT and ASBCD-HTP) run on 20 threads.

We set the step-size $\eta = \frac{1}{48\rho_{\hat{s}}}$ and batch size $|\mathcal{B}| = 1$ during our experiments. The upper bound is then computed according to Theorem 1. Under this condition, we obtain the result, as is shown in Figure 10. As we can see that the upper bound converges faster at first and then it becomes slow since the statistical error becomes the main factor. The convergence performance of SBCD-HTP is always bounded below the black line, which demonstrates the correctness of our theorem.

## F.8 LINEAR SVM

In this subsection, we consider the sparsity-constrained $L_2$-SVM problem (i.e., $\min_w \frac{1}{2n} \sum_{i=1}^{n} \left( \max\{0, 1 - y^{(i)} w^\top x^{(i)}\} \right)^2 + \frac{\lambda}{2} \|w\|^2$, subject to $\|w\|_0 \leq s$), where $y^{(i)}$ is the known label and $x^{(i)}$ is the given sample. We fix the regularization parameter $\lambda = 10^{-4}$. The other parameters are also same to the above experiments. The objective value gap and running time of the baselines are provided in Figure 11. We can see that we also make a similar observation as from the previous results. SBCD-HT performs the best among these considered baselines.

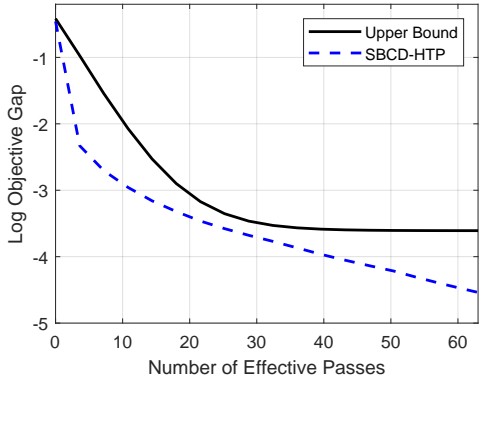

(a) $n = 5000, d = 10000, s = 500, s^* = 500$

Figure 10: The upper bound of Theorem 1 vs the convergence performance.

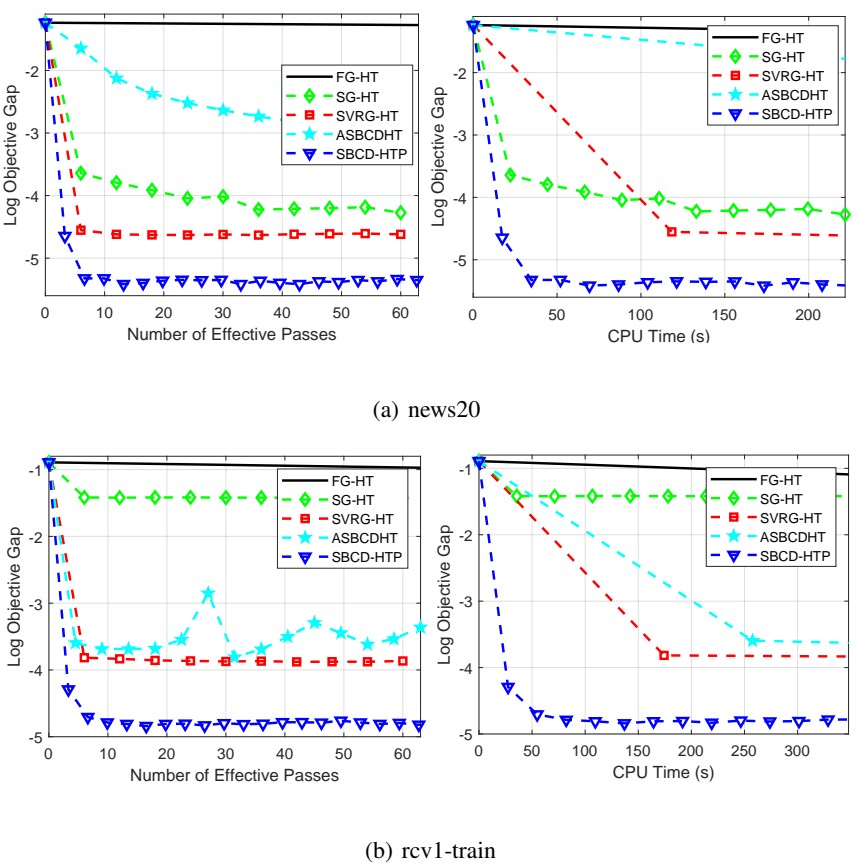

(a) news20

(b) rcv1-train

Figure 11: Comparison of FG-HT, SG-HT, SVRH-HT, ASBCDHT and SBCD-HTP for solving sparse SVM problems.

