# OpenReview forum: "Efficient High-Dimensional Data Representation Learning via Semi-Stochastic Block Coordinate Descent Methods"
_ICLR.cc/2020/Conference — Reject_

### Official Review · AnonReviewer2 · 2019-10-23
**Official Blind Review #2**

**Rating:** 3

**Review:**

This paper proposes a semi-stochastic block coordinate descent hard thresholding pursuit (SBCD-HTP) algorithm for solving l0 sparsity-constrained minimization with decomposable convex objective. The key idea is to introduce BCD into SVRG type of algorithm to speed up the convergence as well as the hard thresholding operation. The paper is well written and easy to follow. The theoretical analysis is strong, I think. However, my main concern of the paper is the experimental part.

1. The test functions are simple (i.e. logistic and linear regression), and the data sets are simple as well. I'd like to see more experiments using more complex problems with more realistic data.

2. There are two stronger assumptions in the paper, but they are never verified in the experiments. What else functions are following such functions? Any example?

3. There are no validation experiments for Thm. 1. Are all the assumptions mild in practice? Is it possible to plot the upper bound in Eq. 4?

4. I do not understand the usage of Def. 1.


**Experience Assessment:**

I have published one or two papers in this area.

**Review Assessment: Checking Correctness Of Derivations And Theory:**

I assessed the sensibility of the derivations and theory.

**Review Assessment: Checking Correctness Of Experiments:**

I assessed the sensibility of the experiments.

**Review Assessment: Thoroughness In Paper Reading:**

I read the paper at least twice and used my best judgement in assessing the paper.

---

> ### Author Response · Authors · 2019-11-14
> **Response to Reviewer #2**
>
> Thank you for positive comments. We address your concerns as follows.
>
> Q1：The test functions are simple (i.e. logistic and linear regression), and the data sets are simple as well. I'd like to see more experiments using more complex problems with more realistic data.
> R1: We have added more experimental results for solving the linear support vector machine (SVM) problem in the revised manuscript (please see Figure 11 in the revised manuscript). W
>
> e have also conductedWe make many experiments on five real-world (realistic) datasets including rcv1-test, rcv1-train, real-sim, news20, and , E2006-TFIDF. In addition, Many many experimental results (e.g., on face recognition tasks) are have been provided in Appendix (see Section F in Appendix). And all the models and datasets we use include the models and datasets have been used by almost all the state-of-the-art papers.
>
> Q2：There are two stronger assumptions in the paper, but they are never verified in the experiments. What else functions are following such functions? Any example?
>
> R2: In fact, the two strong assumptions (i.e., Restricted Strong Convexity (RSC) and Restricted Strong Smoothness (RSS)) are commonly used in the literature of hard thresholding algorithms (such as Zhou et al. [2018], Yuan et al. [2014] and so on), which is are also stated in Section 3 in our main paper. Beside the linear / logistic regression functions, the multi-classes of softmax regression and linear SVM functions are also included in this field.
>
> Pan Zhou, Xiao-Tong Yuan, Jiashi Feng. Efficient Stochastic Gradient Hard Thresholding, NeurIPS, 2018.
>
> Xiao-Tong Yuan, Ping Li, Tong Zhang. Gradient Hard Thresholding Pursuit for Sparsity-Constrained Optimization, ICML, 2014
>
> Q3: There are no validation experiments for Thm. 1. Are all the assumptions mild in practice? Is it possible to plot the upper bound in Eq. 4?
>
> R3：For the convergence rate of Theorem 1, we can see that the proposed algorithms achieve the best-known theoretical convergence result for the sparsity-constrained non-convex problem, and all the experimental results in our paper show that our algorithms have better performance than the state-of-the-art methods in terms of both effective pass and CPU time. To address your concern, we have added the comparison between the theoretical upper bound and the actual performance in the Appendix in the revised manuscript (please see Figure 10). All the experimental results show the correctness of our algorithms and theoretical analysis.
>
> Q4: I do not understand the usage of Def. 1.
>
> R4: We use Definition 1 is used to define a hard thresholding complexity, as in Zhou et al [2018]. Based on this definition, we can evaluate the complexity of hard thresholding operations in theory, as in few several existing work such as ((Zhou et al., 2018)).
>
> Pan Zhou, Xiao-Tong Yuan, Jiashi Feng, Efficient Stochastic Gradient Hard Thresholding. NeurIPS, 2018.

---

### Official Review · AnonReviewer1 · 2019-10-27
**Official Blind Review #1**

**Rating:** 3

**Review:**

This paper introduces a new optimization algorithm for convex functions with sparsity constraints, based on hard thresholding to enforce sparsity. It is argued that applying hard thresholding in every step is the computational bottleneck of previous works based on hard thresholding, and the paper proposes a variant that alleviates previous works’ convergence problems. This is achieved by having an inner and outer loop in which the hard thresholding is applied only in the outer loop, and in the inner loop block coordinate descent is used  (ie. applying updates to only a subset of the solution’s variables). The paper shows that the convergence bounds of the algorithms compares favourably to previous hard thresholding algorithms convergence bounds, and empirical results show the effectiveness of the algorithm compared to previous ones.

It is always exciting to see improvements in optimization algorithms as they can lead to improvements in many different problems. Yet, I have the following concerns:
-The algorithm is strongly based on previous optimization algorithms as the main difference with them is applying the hard thresholding less frequently. This sounds incremental and I think the paper could emphasize much more why this idea is a significant advancement over previous works.
-The paper could make a better job motivating the sparsity constraints, as it cites papers from 10 years ago which are not representative of state-of-the-art.
-The paper focuses on optimization time but not on generalization. Note that converging faster to the objective does not imply better generalization. I would have expected at least a discussion about this in the paper. Does the algorithm lead to better generalization accuracy or worse?
-The proof contains an assumption that it is unclear that has been explained in the paper. In the supplementary material, after eq. 25, the factor sigma is being introduced and the paper says “we assume that there exists a constant factor sigma making this inequality true”.
-Definition 1 does not really define hard thresholding.


**Experience Assessment:**

I have read many papers in this area.

**Review Assessment: Checking Correctness Of Derivations And Theory:**

I assessed the sensibility of the derivations and theory.

**Review Assessment: Checking Correctness Of Experiments:**

I assessed the sensibility of the experiments.

**Review Assessment: Thoroughness In Paper Reading:**

I read the paper at least twice and used my best judgement in assessing the paper.

---

> ### Author Response · Authors · 2019-11-14
> **Response to Reviewer #1**
>
> Thank you for your positive comments. We address your concerns as follows.
>
> Q1:The algorithm is strongly based on previous optimization algorithms as the main difference with them is applying the hard thresholding less frequently. This sounds incremental and I think the paper could emphasize much more why this idea is a significant advancement over previous works.
>
> R1: In our theoretical analysis, we can see that the hard thresholding operation used in each inner- loop, which is very time-consuming, usually O(dlog(d)). However, we reduce the hard thresholding complexity from linear dependence on kappa_shat (i.e., the restricted condition number) to an independent result (i.e.,  O(kappa_shatlog(\frac{1}{\epslion}))  vs, . O(log(\frac{1}{\epslion})) ). This is a significant improvement for reducing the whole complexity of hard thresholding algorithms. Moreover, we have improved the required value of sparsity levels from the quadratic dependence on kappa_shat to a linear dependence result. It means that we can choose more flexible in the sparsity level to obtain an approximate solution of the sparsity-constrained non-convex problem.
>
> Q2: The paper could make a better job motivating the sparsity constraints, as it cites papers from 10 years ago which are not representative of state-of-the-art.
>
> R2: Thanks for your suggestion. In the previous manuscript, we have cited some state-of-the-art work for solving the sparsity-constrained problem, such as (Zhou et al, 2018) and (Chen et al., 2017). To address your concern, we have added more references, e.g., the state-of-the-art papers 10 years ago in the revised manuscript, such as (Yang et al., 2010), (Zhang et al., 2011).
>
> Zhou P, Yuan X, Feng J. Efficient stochastic gradient hard thresholding. NeurIPS, 2018.
> Chen J, Gu Q. Fast newton hard thresholding pursuit for sparsity constrained nonconvex optimization.SIGKDD, 2017.
> Yang J, Wright J, Huang T S, et al. Image super-resolution via sparse representation. IEEE transactions on image processing, 2010, 19(11): 2861-2873.
> Zhang L, Yang M, Feng X. Sparse representation or collaborative representation: Which helps face recognition ICCV, 2011: 471-478.
>
> Q3: The paper focuses on optimization time but not on generalization. Note that converging faster to the objective does not imply better generalization. I would have expected at least a discussion about this in the paper. Does the algorithm lead to better generalization accuracy or worse?
>
> R3: In fact, we have conducted some experiments for the generalization ability in the Appendix on the face recognition task in the Appendix. All the results show that our SBCD-HT algorithm has a higher testing accuracy than the state-of-the-art methods (e.g., SVRG-HT) on the Extended Yale B dataset (please see Figure 5)same test set.
>
>
> Q4: The proof contains an assumption that it is unclear that has been explained in the paper. In the supplementary material, after eq. 25, the factor sigma is being introduced and the paper says “we assume that there exists a constant factor sigma making this inequality true”
>
> R4: Actually, in Eq. (25), we can see that the right-hand side of Eq. (25) is less than F(w^r)-F(w^m), since the L2-norm is always positive. Thus, the factor sigma could be used to describe the tight bound of this equation. This is not a strong assumption in the analysis and it can be satisfied by using a small sigma.
>
> Q5：Definition 1 does not really define hard thresholding.
> R5: In fact, Definition 1 is to define not a hard thresholding operation but the hard thresholding complexity, not a hard thresholding operation. Definition 1 is the same as the definition of the hard thresholding complexity as in (Zhou et al., 2018).
>
> Pan Zhou, Xiao-Tong Yuan, Jiashi Feng. Efficient Stochastic Gradient Hard Thresholding, NeurIPS, 2018.

---

### Official Review · AnonReviewer3 · 2019-11-02
**Official Blind Review #3**

**Rating:** 6

**Review:**

===== Update after author response

Thanks for the clarifications and edits in the paper.

I recommend acceptance of the paper.

Other comments:
Definition 1 in the updated version is still too vague ("difference of what?" -- function values? distance in norm between iterates?) -- this should be clarified.

========

This paper considers the problem of sparsity-constrained ERM and asks whether one can design a variant of the stochastic hard thresholding approaches where the hard-thresholding complexity does not depend on a (sparsity dependent) condition number, unlike all previous approaches (Table 1). It proposes a method which combines SVRG-type variance reduction, with block-coordinate updates, leaving the hard thresholding operation outside the inner loop, to accomplish this goal. It provides a convergence analysis which significantly improves the previous best rates (by having both the sparsity level shat which is significantly lower (kappa_shat vs. kappa_stilde^2) as well as a condition number independent hard thresholding complexity (Table 1). An asynchronous and sparse (in the features) variant is also proposed, with even better complexity. Some standard experiments on sparse linear regression and sparse logistic regression is presented showing an improvement in both number of iterations as well as CPU time.

I think the clarity of the paper should be quite improved (see detailed comment), hence why I think the paper is borderline, but I am leaning towards an accept given the significant theoretical improvements over the past literature (and positive empirical results), even though the algorithmic suggestion is somewhat incremental.

The proposed Algorithm 1 seems very close to the one of Chen & Gu (2016), the paper should be more clear about this. There seems to be mainly two changes: a) extending the support projection of the gradient to the union of the sampled block with the one of the support of the reference parameter wtilde (vs. just the sampled block in Chen & Gu (2016) and b) moving the hard-thresholding iteration outside of the SVRG-inner loop. These small tweaks to the algorithm yield a significant theoretical improvement, though.

== Detailed comments ==

Clarity: the number of long abbreviations with only one letter change make it hard to follow the different algorithms; perhaps a better more differentiating naming scheme could be used. Moreover, I think more background on the sparse optimization setup should be provided in the introduction or at least in the preliminaries, as I do not think the wider ICLR community is very familiar with it (in particular, no cited paper was at ICLR). For example, define early the separation in optimization error and statistical error; and point out that F(w_t) might even be lower than F(w*) as the sparsity threshold s might be much higher than s*. This will make Table 1 more concrete and less abstract for people who not are not yet experts on this particular analysis framework.

- Table 1: I would suggest to put the rate for S2BCD-HTP instead on the last row and mention instead that the rate for ASBCD is similar under conditions on the delay; as it is interesting to already have a better gradient complexity for S2BCD vs. SBCD.

** Questions:  **
1) In Corollary 1, how is the gradient oracle complexity defined or computed? And more specifically, how does one compare fairly the cost of doing a gradient update in Algorithm 1 on the *bigger set* S = Gtilde U G_jt vs. just G_jt for the Chen & Gu ASBCD algorithm? Is this accounted in the computation?

2) In Figure 1, which "minimum" is referred to and how is it found? I suspect it is not F(w*) (as it could be higher than F(w_t)), i.e. it is *not* the minimum of (1) with s*. One natural guess is that it might be min_w F(w) s.t. ||w||_0 <= s, though I do not see any guarantee in the main paper that running the algorithm would make F(w_t) converge to such a value (i.e. all we know from Thm 1 is that F(w_t) might be within O(||nabla_Itilde F(w*)||^2) of F(w*) ultimately. Please explain and clarify!

== Potential improvement ==

The current result in Theorem 1, which is building on a similar proof technique as the original SVRG paper, has the annoying property of requiring the knowledge of the condition number in setting the size of the inner loop iteration. I suspect that this is an artifact of using an outdated version of the SVRG algorithm. This has been solved since then by considering a "loopless" version of SVRG which implicitly defines the size of the inner loop in a random manner using a quantity *which does not depend on the condition number*. This was proposed first by Hofmann et al. [2015], and then re-used by Lei & Jordan [2016] and more recently by Kovalev et al. [2019] e.g. Note that Leblond et al. (2017) that you cited profusely also used this variant of SVRG. I suspect that this technique could be re-used in your case to obtain a similar result with a loopless variant (which also gives cleaner complexity results). (Though I only skimmed through your proof.)

Caveat: the sensibility of the theory in the main paper seems reasonable, but I did not check the proofs in the appendix.

= References:
- Hofmann et al. [2015]: Variance Reduced Stochastic Gradient Descent with Neighbors, Thomas Hofmann, Aurelien Lucchi, Simon Lacoste-Julien and Brian McWilliams, NeurIPS 2015
- Lei & Jordan [2016]: Less than a Single Pass: Stochastically Controlled Stochastic Gradient Method, Lihua Lei and Michael I. Jordan, AISTATS 2016
- Kovalev et al. [2019]: Don't Jump Through Hoops and Remove Those Loops: SVRG and Katyusha are Better Without the Outer Loop, Dmitry Kovalev, Samuel Horvath and Peter Richtarik, arXiv 2019


**Experience Assessment:**

I have published in this field for several years.

**Review Assessment: Checking Correctness Of Derivations And Theory:**

I assessed the sensibility of the derivations and theory.

**Review Assessment: Checking Correctness Of Experiments:**

I assessed the sensibility of the experiments.

**Review Assessment: Thoroughness In Paper Reading:**

I read the paper at least twice and used my best judgement in assessing the paper.

---

> ### Author Response · Authors · 2019-11-14
> **Response to Reviewer #3**
>
> Thank you for your positive comments. We address your concerns as follows.
>
> Q1: In Corollary 1, how is the gradient oracle complexity defined or computed? And more specifically, how does one compare fairly the cost of doing a gradient update in Algorithm 1 on the *bigger set* S = Gtilde U G_jt vs. just G_jt for the Chen & Gu ASBCD algorithm? Is this accounted in the computation?
>
> R1: In the computation of gradient oracle complexity, we omit the smaller set Gtilde since we cannot guarantee there is not overlap between Gtilde and G_jt and taking the dimension d into consideration, and thus the influence of Gtilde maybe actually small. We have shown the worst condition of this gradient complexity in the revised manuscript, and also made some discussions about this issue in the rRemark 1 in the revised manuscript.
>
> Q2: In Figure 1, which "minimum" is referred to and how is it found? I suspect it is not F(w*) (as it could be higher than F(w_t)), i.e. it is *not* the minimum of (1) with s*. One natural guess is that it might be min_w F(w) s.t. ||w||_0 <= s, though I do not see any guarantee in the main paper that running the algorithm would make F(w_t) converge to such a value (i.e. all we know from Thm 1 is that F(w_t) might be within O(||nabla_Itilde F(w*)||^2) of F(w*) ultimately. Please explain and clarify!
>
> R2: As in (Zhou et al., 2018) , we run all the algorithms (e.g., FG-HT, SG-HT, SVRG-HT, ASBCDHT, and SBCD-HTP) sufficiently long until \|w^t-w^(t-1)\|/\|w^t\|<10e-6, since there is no ground truth on the real-world datasets. Then we regard the minimum of all the results to the optimal F(w^*). To address your concern, we have clarified this issue at the beginning of Section 6 in the revised manuscript.
>
> Pan Zhou, Xiao-Tong Yuan, Jiashi Feng. Efficient Stochastic Gradient Hard Thresholding, NeurIPS, 2018.

---

### Decision · Program_Chairs · 2019-12-19

**Decision:**

Reject

**Comment:**

All the reviewers reach a consensus to reject the current submission.

In addition, there are two assumptions in the proof which seemed never included in Theorem conditions or verified in typical cases.

1) Between Eq (16) and (17), the authors assumed the 'extended restricted strong convexity’ given by the un-numbered equation.

2) In Eq. (25), the authors assume the existence of \sigma making the inequality true.

However those assumptions are neither explicitly stated in theorem conditions, nor verified for typical cases in applications, e.g. even the square or logistic loss. The authors need to address these assumptions explicitly rather than using them from nowhere.